# Membrane reshaping by micrometric curvature sensitive septin filaments

Alexandre Beber[1,2], Cyntia Taveneau[1,2], Manuela Nania[3], Feng-Ching Tsai[1,2], Aurelie Di Cicco[1,2], Patricia Bassereau[1,2], Daniel Lévy[1,2], João T. Cabral [3], Hervé Isambert[1,2], Stéphanie Mangenot[1,2] & Aurélie Bertin[1,2]

Septins are cytoskeletal filaments that assemble at the inner face of the plasma membrane. They are localized at constriction sites and impact membrane remodeling. We report in vitro tools to examine how yeast septins behave on curved and deformable membranes. Septins reshape the membranes of Giant Unilamellar Vesicles with the formation of periodic spikes, while flattening smaller vesicles. We show that membrane deformations are associated to preferential arrangement of septin filaments on specific curvatures. When binding to bilayers supported on custom-designed periodic wavy patterns displaying positive and negative micrometric radii of curvatures, septin filaments remain straight and perpendicular to the curvature of the convex parts, while bending negatively to follow concave geometries. Based on these results, we propose a theoretical model that describes the deformations and micrometric curvature sensitivity observed in vitro. The model captures the reorganizations of septin filaments throughout cytokinesis in vivo, providing mechanistic insights into cell division.

[1] Laboratoire Physico Chimie Curie, Institut Curie, PSL Research University, CNRS UMR168, 75005 Paris, France. [2] Sorbonne Université, 75005 Paris, France. [3] Department of Chemical Engineering, Imperial College London, London SW7 2AZ, UK. These authors jointly supervised this work: Stéphanie Mangenot, Aurélie Bertin. Correspondence and requests for materials should be addressed to S.M. (email: stephanie.mangenot@curie.fr) or to A.B. (email: aurelie.bertin@curie.fr)

Septin filaments constitute a category of eukaryotic cytoskeletal filaments[1]. Septins are ubiquitous and essential to a variety of cellular processes from cell division[2,3], neuron morphogenesis[4], cell motility[5], or cellular compartmentalization[6–9]. Septins are involved in membrane remodeling processes.

Septins self-assemble into linear palindromic complexes (Supplementary Fig. 1). The mitotic *Saccharomyces cerevisiae* complex gathers five septin subunits[10,11] to form a 32 nm long rod-like complex. The minimal septin oligomers, at low ionic strengths, self-assemble into micrometric long non-polarpaired filaments (Supplementary Fig. 1). In vitro studies have shown that septins organize specifically on PI(4,5)P2-containing membranes[12,13].

In vivo, *Saccharomyces cerevisiae* septins assemble at the bud neck of dividing cells[14,15]. They bind to the inner plasma membrane through specific interactions with PI(4,5)P2[12]. At the cleavage furrow, the membrane displays a negative Gaussian curvature, horse saddle geometry, with both a positive and negative curvature. Evidences show that septin filaments globally reorient by 90° at the onset of cytokinesis in budding yeast[16,17]. Initially aligned along the mother–bud axis, septins rearrange in two distinct rings aligned circumferentially around the bud neck on both sides of the contractile acto-myosin ring.

Curvature sensing is involved in cell and organelle morphology. Four different mechanisms have been proposed to generate and sense membrane curvature (for a review see Callan-Jones and Bassereau[18–21]). In all of these processes, the sensed curvatures are within 1 nm$^{-1}$ range. However, septins localize at curvatures of 1 μm$^{-1}$ as seen around the bud neck during cell division. It is thus crucial to understand how filaments can assemble, grow and organize on micrometric radius of curvatures[22].

Studies using silica beads of defined diameters coated with supported lipid bilayers have shown that septins sense and have a preference for 2 μm$^{-1}$ curvatures[22]. Using spheres with only positive Gaussian curvatures does not allow to study how septin filaments sense negative curvatures or to describe their organization.

In the present work, we design an in vitro system based on lipid bilayers supported on custom-designed periodic wavy patterns to perform a comparative analysis of the preference of septins for negative rather than positive curvature, in the micrometer range. Measurements with giant unilamellar vesicles (GUVs) show that septins are able to negatively curve membranes on micrometer scales and that septins barely affect the mechanical properties of membranes. Based on our in vitro observations, we propose a simple theory relying on the persistence length and the adhesion energy of septin filaments to account for these deformations and the curvature preference of septins. Besides, the major septin rearrangements during cell division, starting with parallel septin filaments along the bud axis and ending with circumferential rings at the bud neck before cytokinesis is accounted in our model. We thereby reveal mechanistic aspects of cell division.

## Results

**Septins reduce the apparent projected area of vesicles.** To investigate how septins can reshape and/or alter the mechanical properties of biomimetic membranes, we have performed a micropipette experiment[23]. Septins organize into palindromic rods which further self-assemble into non-polarpaired filaments (Supplementary Fig. 1). A Giant Unilamellar Vesicle doped with 8% PI(4,5)P2 was held by a micropipette at a fixed tension, creating a stable tongue within the pipette (Fig. 1a). All the errors from this report are calculated using standard deviations. Septins were then injected with a second micropipette

at concentrations up to 650 nM, leading to a local bulk septin concentration around 300 nM. As shown in Fig. 1b, during septin addition, at a fixed tension, the tongue length decreased significantly with a gradual interaction of septins with the vesicle. It implies that the apparent area, related to the tongue length, decreased simultaneously with the addition of septins on the vesicle. Fig. 1c presents an example of volume and apparent area variations with the septin density on the membrane. In all of these experiments, the volume of the vesicles remained constant, which excludes the possibility of osmotic effects. The mean relative variation area for all vesicles as a function of bound proteins is presented in Fig. 1d ($N = 54$ GUVs). On average, we observed a reduction by about 2% of the apparent area with a septin density of 6000 septins μm$^{-2}$. The decrease in apparent area could result either from the alteration of the mechanical membrane properties or from reshaping of the membrane[24].

**Septins do not significantly alter membrane stiffness.** To investigate whether septins affect the mechanical properties of lipid bilayers, we have measured the bending modulus, κ, of GUVs[25] (for details, see material and methods and Supplementary Fig. 2). The bending modulus κ was unaffected by the presence of septins. It is equal to $10.5 \pm 0.5\, k_b T$ ($N = 37$) and $9.3 \pm 0.7$ ($N = 32$) (average values) in the presence and absence of septins, respectively (Fig. 1f). In the absence of proteins, the value of the bending modulus, κ, is in perfect agreement with values previously published[26]. To probe the response of membranes at higher tensions, we have determined the stretching modulus. The stretching modulus in the presence of septins ($42 \pm 8$ mN m$^{-1}$) is slightly lower than the stretching modulus of naked vesicles ($65 \pm 9$ mN m$^{-1}$) (Supplementary Fig. 3). In the presence of septins, external stretching of the membrane could trigger septin insertion between lipids end thus lower the stretching modulus. Hence, we considered that septins hardly affect the mechanical properties of membranes.

**Septins deform GUVs in 3D and induce the formation of spikes.** To address the ability of septins to deform membranes in three dimensions, we have incubated GUVs-containing PI(4,5)P2 with solutions of septins for one hour, at concentrations up to 650 nM. In contrast with the pipette experiments, GUVs were not stretched and thus not spherical anymore. In the lower concentration range (about 200 nM) and at low ionic strength conditions (75 mM NaCl, 10 mM Tris pH 7.6) where septins polymerize, GUVs did not exhibit a smooth spherical shape. Instead, GUVs homogeneously covered with septins appeared facetted, and displayed sharp angles (Fig. 2a). Septins localized on vesicles either homogeneously (Fig. 2) or by assembling into bundles (Supplementary Fig. 4). When septin bundles were visible at the surface of the vesicles, the vesicle shape was unaffected (Supplementary Fig. 4). Those bundles were probably selfassembled in solution prior to their interaction with the vesicles. At higher protein concentration and with a homogeneous protein coverage, the deformations were much more pronounced and spikes were visible at the surface of the vesicles (Fig. 2b). At 600 nM in bulk, one-third of the GUVs displayed spiky protrusions. The periodicity, the radius of curvature and the amplitude of the protrusions were measured (Fig. 2c). The mean amplitude of the spikes was measured to be $0.9 \pm 0.2$ μm and their periodicity $3.9 \pm 1.4$ μm ($N = 43$ vesicles). The inwards curvature (concave) of the membrane deformation in between consecutive protrusions was $1.1 \pm 0.2$ μm$^{-1}$ ($N = 43$) (Fig. 2d). Hence, the deformations were well defined and regular. Furthermore, the periodicity of the spikes was independent of the

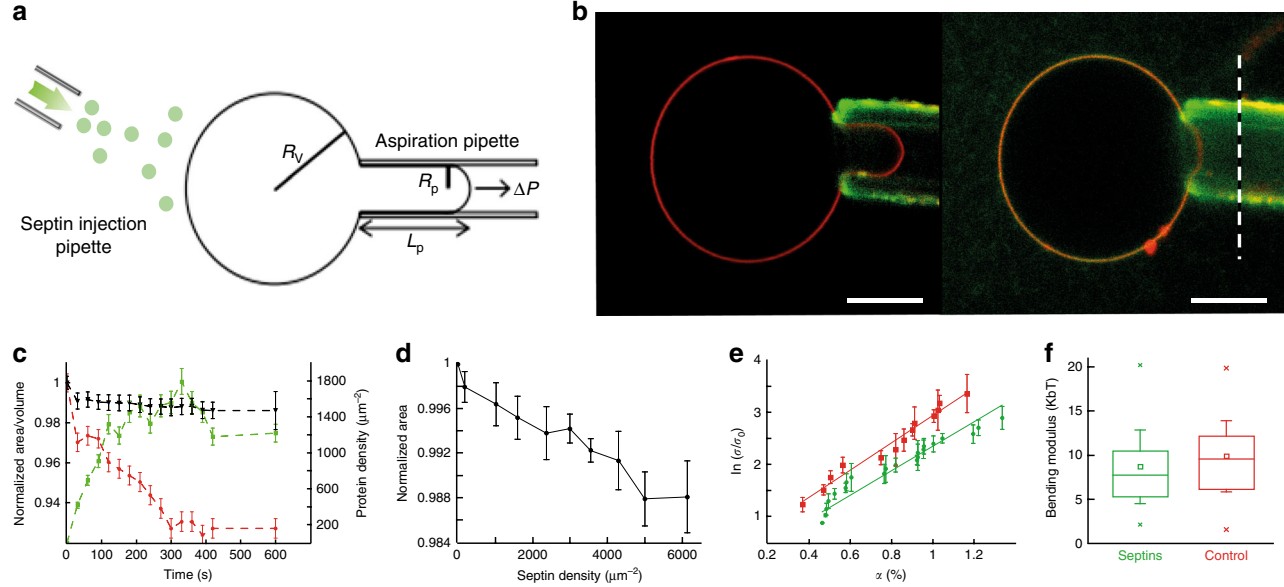

**Fig. 1** Micropipette experiment and GUV mechanics. **a** Illustration of a typical micropipette experiment where $R_v$, $R_p$, $L_p$, and $\Delta P$, are, respectively, the vesicle radius, the pipette radius, the tongue length, and the applied pressure. **b** Confocal images of a GUV held by a micropipette at constant tension, where lipids are labeled in red and septins in green. The first image corresponds to $t = 0$, before addition of septins and the second image to $t = 10$ min; the dashed line represents the position of the tongue length at $t = 0$. Scale bar = 10 μm. **c** Typical curve representing the relative change over time in area (red squares) and volume (black triangles), as a response to the binding of septins on a GUV held by a micropipette. The red squares correspond to the density of bound septins. Error bars represent s.d. (standard deviations). **d** Mean curve of the change of area as a function of bound septin density ($N = 56$). **e** Mean curves of the change of area as a function of applied tension without septins (red dots, $N = 37$) and at fixed septin density $d = 3000$–$6000$ μm$^{-1}$ (green squares, $N = 27$). Dotted lines represent linear fits. Error bars represent s.d. **f** Box plot of measured bending moduli. Boxes represent the 25–75% range, crosses are min and max values, whiskers are standard deviations and squares are mean values

size of the vesicles, for GUV diameters varying from 5 to 40 μm (Supplementary Fig. 5).

At higher ionic strengths (above 150 mM NaCl), which impede septin polymerization in solution, septins still bound to and deformed GUVs ($N = 23$) (Fig. 2e). This observation demonstrates that the interaction of septins with GUVs does not solely depend on electrostatics but relies on more specific bonds with PI (4,5)P2.

To assess whether a steady state was reached in our experiments, we have incubated septins and vesicles overnight. Instead of multiple spikes, we observed extremely deformed vesicles with a peculiar star shape with only a few protrusions (Supplementary Fig. 6).

The reduction of the GUVs apparent area observed in Fig. 1d probably results from a local membrane reshaping in 3D. Membrane reshaping, such as macroscopic tubulation, leads to a reduction of the projected area of GUV[24].

To account for the deformations of GUVs induced by septins, we have developed a minimal theoretical model. The interaction of septin filaments with a curved membrane is described below. It results from the competition between two energy terms: a negative binding energy, favoring filament adsorption, and a positive bending energy, weakening filament adsorption. As schematically shown in Fig. 2f, considering an incompressible constant filament volume, the surface of contact between the membrane and a filament actually depends on the surface curvature, with a slightly longer adhesion surface of a filament on a negatively curved surface. This is the central feature of this model, which accounts for the preference of septin filaments for negatively curved membranes, as demonstrated below. The model does not include the actual polymerization process of septins and assumes that they are already filamentous.

The free energy per unit length $\Delta g_1^0$ of a single septin filament bound to a flat membrane ($\Delta g_1^0 < 0$) can be expressed as,

$$\Delta g_1^0 = (k_b T \ln K_d) \frac{1}{8a_1} \tag{1}$$

where $k_b T$ is the thermal energy, $a_1 \approx 4$ nm the size of a septin monomer, and $K_d = 88 \pm 6$ nM is the dissociation constant per octamer, measured on a flat surface covered by a supported bilayer (Methods, Supplementary Fig. 9). Entropic terms are included in the measured free energy parameters.

Assuming a simple continuous interaction model, the free energy per unit length of a single septin filament bound to a curved membrane substrate of curvature $c = \frac{1}{R}$ is expressed as,

$$\Delta g_1(c) = \Delta g_1^0 \left(1 - \frac{1}{2}a_1 c\right) + \frac{1}{2}k_b T L_{p1} c^2 \tag{2}$$

where the term $-\frac{1}{2}a_1 c$ corresponds to the change of contact area due to the curvature, which increases for negative curvature and decreases for positive curvature, following the convention that the outside (respectively inside) of a sphere corresponds to a positive (respectively negative) curvature. The second term, quadratic in curvature, corresponds to the bending energy with $L_{p1}$ the persistence length of a single filament.

Similarly, the free energy per unit length of a septin bundle of ($n$) filaments bound to a curved membrane substrate is,

$$\Delta g_n(c) = \Delta g_n^0 \left(1 - \frac{1}{2}a_n c\right) + \frac{1}{2}k_b T L_{pn} c^2 \tag{3}$$

where $a_n$, $L_{pn}$, and $\Delta g_n^0$ are the width, the persistence length and the free energy per unit length of a septin multiple filament bound to a flat membrane. Equation (3) exhibits a minimum free

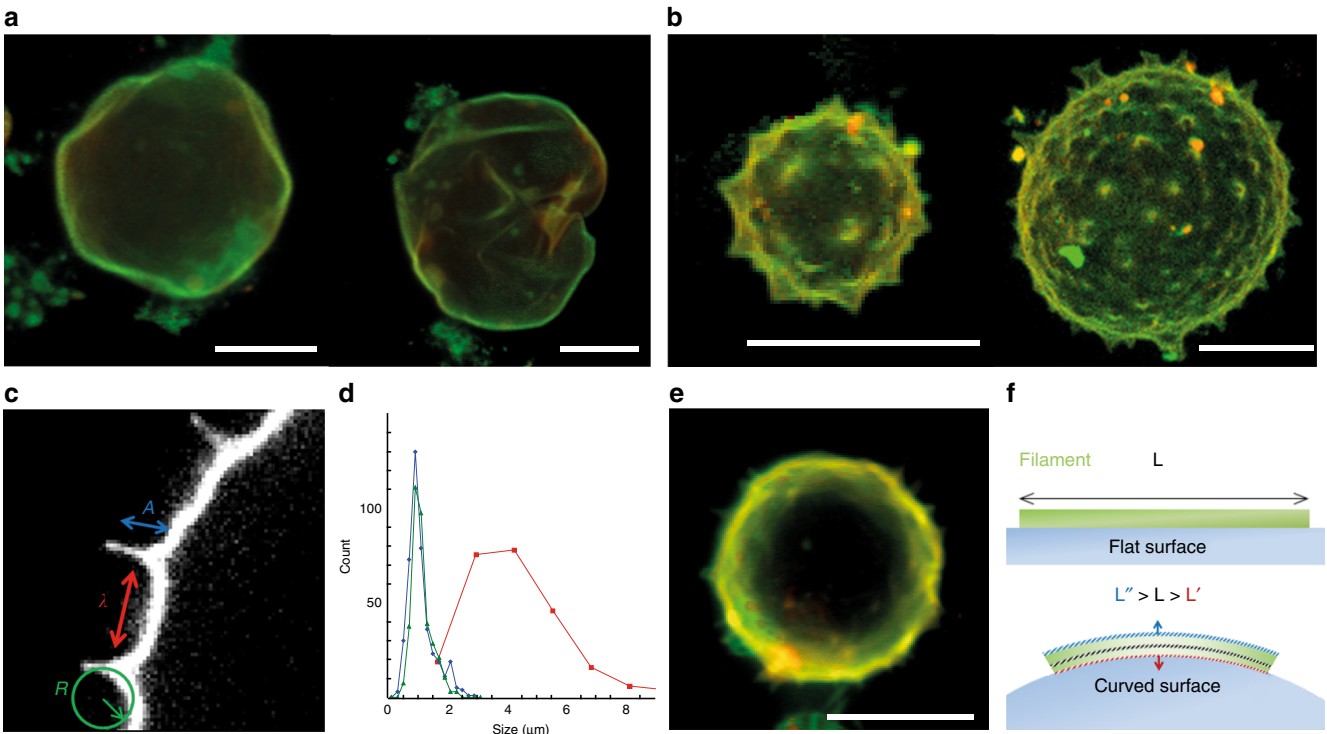

**Fig. 2** Septins induce deformations on GUVs. **a** 3D reconstructions of confocal spinning disk images of GUVs in a solution of 200 nM septins (red = lipids, green = septins). GUVs appear deformed and undulated. **b** 3D reconstructions of confocal spinning disk images of GUVs in a solution of 600 nM septins. Spikes appear on the surface of the GUVs. **c** Confocal image of a spiky GUV in a solution of 600 nM septins. The different parameters $R$ (radius of curvature), $A$ (height) and $\lambda$ (distance) of the spike are highlighted. **d** Distribution of parameters height, $A$, (blue); radius of curvature, $R$ (green); and distance, $\lambda$ (red) of the spike on GUVs ($N_{\text{vesicles}} = 35$). **e** 3D reconstruction of spinning disk image of a GUV in a solution of 600 nM septins at high salt concentration (300 mM NaCl). **f** Sketch of a septin filament bound to a curved membrane. At a constant volume, a bent filament shows a greater surface towards curved membrane. All scale bars are 10 µm

energy corresponding to an optimum negative curvature of septin filaments bound to a membrane,

$$c_n^* = \frac{1}{R_n^*} = \frac{\Delta g_n^0 a_n}{2 k_b T L_{pn}} \qquad (4)$$

Assuming a (roughly) circular section of multiple septin filaments, $a_n$, $\Delta g_n^0$, $c_n^*$, $R_n^*$ are expected to scale as,

$$a_n \approx a_1 n^{\frac{1}{2}} \qquad (5)$$

$$L_{pn} \approx L_{p1} n^2 \qquad (6)$$

$$\Delta g_n^0 \approx \Delta g_1^0 n^{\frac{1}{2}} \qquad (7)$$

$$c_n^* \approx c_1^* n^{-1} \qquad (8)$$

$$R_n^* \approx R_1^* n^1 \qquad (9)$$

In solution, budding yeast–septins can only assemble into paired filaments[12]. We have experimentally measured the persistence length of budding yeast–septin-paired filaments $L_{p2}$ by analyzing TIRF movies of septin filaments freely fluctuating on a passivated glass surface (300 images, Supplementary Fig. 10A and B, see Methods section for details[27]) and found $L_{p2} \approx 8\,\mu m$. Then, Eq. (6) provides an estimate of the persistence length of a single filament $L_{p1} \approx \frac{L_{p2}}{4} \approx 2\,\mu m$. The bending energy of septins

is included in the model and corresponds to the terms with the persistence length, $L_p$, in the equations.

As previously shown[12], budding yeast–septins can bind to biomimetic membranes as single filaments. As a consequence, in these conditions, the optimum curvature of a single septin filament bound to a membrane is estimated to be $c_1^* \approx \frac{\Delta g_1^0 a_1}{2 k_b T L_{p1}} \approx -1.4\,\mu m^{-1}$, which is in good agreement with the observations of spontaneous deformation of GUVs due to the adsorption of septin filaments outside of the vesicles, Fig. 2c.

**Septins deform large unilamellar vesicles**. To visualize individual septin filaments, optical microscopy remains limited, in terms of resolution. To circumvent this issue, we have used cryo-electron microscopy and tomography, which enables the visualization of the 3D deformations of membranes and septin filaments[28]. Besides, we could visualize deformations induced by septins on smaller vesicles and thus on higher curvatures than curvatures accessible with GUVs. Large unilamellar vesicles (LUVs) of variable sizes ranging from 50 nm to 1 µm in diameter were obtained and were deformable (see control, Fig. 3a, left panel, insert 1 and Supplementary Fig 7A). Septins do not interact with the smallest vesicles. About 50% were covered with septins as shown in 2D images (Fig. 3a, stars). From images collected in transmission, we observed a gallery of membrane deformations and disruptions from flattening (Fig. 3a, left, and right panel, and Supplementary Fig. 7D), protrusions (Fig. 3a, left panel and Supplementary Fig. 7D), membrane tears and ruptures (Fig. 3a, middle panel, arrow). Septin filaments assemble into individual filaments, networks of orthogonal filaments (Fig. 3a left panel and 3b) or flat sheets of parallel filaments (Fig. 3a,

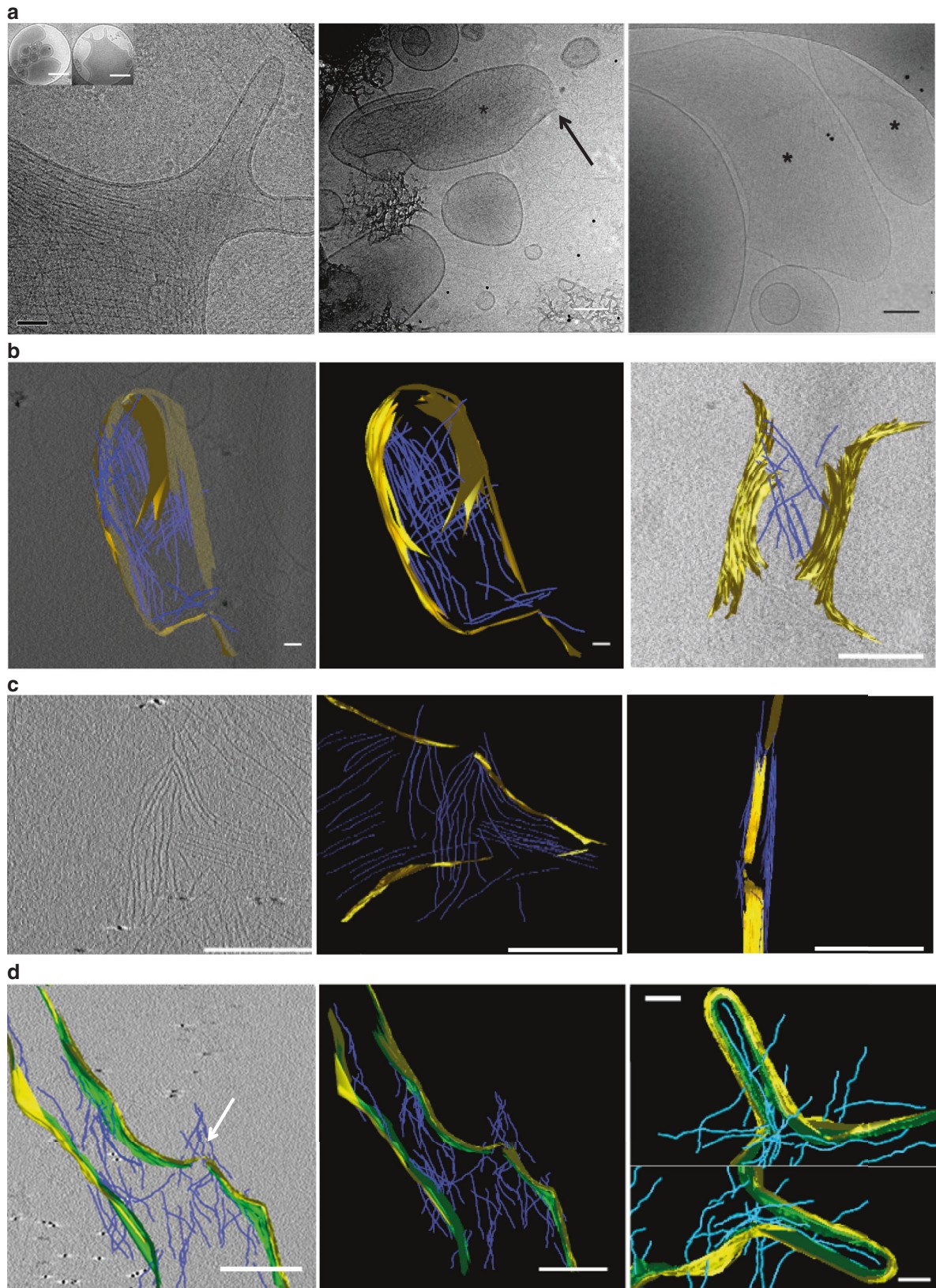

middle and right panel). To discriminate whether filaments are bound to the top or bottom of vesicles and visualize the membrane deformations in 3D, we performed cryo-electron tomography (Supplementary Movies 1 and 2). Twenty cryo-tomograms were collected. Tomograms show that vesicles are flattened to accommodate septin filaments that align parallel to one another (Fig. 3b, c, Supplementary Movies 1 and 2). As visualized in 2D projections (Fig. 3, right panel), naked vesicles remain perfectly round shaped (Supplementary Movies 3 and 4), whereas decorated vesicles display peculiar morphologies. Indeed sharp angles can be pinpointed on the contour of those vesicles (arrows).

**Fig. 3** Cryo-electron microscopy of septins bound to LUVs. **a** Cryo-EM images of septins bound to vesicles. Stars indicate vesicles with septins bound. Left: flattened vesicles with protrusions. The arrow points to septin networks of filaments. Insert 1: Control without septins, insert 2: After septin addition. Middle: example of a ruptured vesicle. The arrows point at the broken membrane. Right: vesicle covered with an array of parallel filaments. Scale bars = 50 nm, inserts: circular holes of 1 μm in diameter. **b** Electron cryo-tomography: segmentation within cryo-tomograms of septins bound to vesicles. The membrane is highlighted in yellow. Filaments are modeled in blue. Middle and left image correspond to a vesicle covered with septin filaments. Right image: deformed membrane covered with filaments. Scale bars = 200 nm. **c** Cryo-tomography of septin filaments around protrusions. Left: slice within a tomogram with the segmentation highlighted in the middle panel. Right: filaments wrapping around a protrusion. Scale bars = 200 nm. **d** Cryo-tomography of septins rupturing vesicles. Left (slice in a tomogram) and middle (segmentation only): filaments breaking the membrane. Scale bars = 200 nm. Right: filaments inside a vesicle aligning within a protrusion. Scale bars = 100 nm

The diameter of the filaments (about 3 nm) corresponds to a single septin filament. The spacing between individual filaments is $14.3 \pm 3$ nm ($N = 195$ filaments, 19 vesicles, 5 experiments). This distance agrees with the spacing imposed by coiled coils in between paired filaments. This suggests that coiled coils extend alternatively on both sides of septin rods to connect septin filaments laterally in a regular array of parallel filaments (Supplementary Fig. 1B). Hence, on vesicles, septins obviously bind as single filaments connected through coiled coils. We indeed assumed for the calculation above (Eqs. (1) and (2)) that single filaments were bound to GUVs. In addition, the behavior of septins on membrane protrusions could be analyzed. On Fig. 3c, Supplementary Fig. 8 and Movie S1, slices in tomogram show that septins do not organize randomly around a protrusion. Indeed, the filaments align along the axis of the bud. Septins also rupture vesicles (Fig. 3d, left panel, arrow). Consequently, filaments can be visualized inside vesicles as displayed in Fig. 3d (right panel) and binding alongside a bud. Overall, these observations demonstrate that septins deform, constrain, and flatten small positively curved vesicles (diameters from 100 nm to 1 μm), suggesting that septins deform membranes to avoid positive curvatures, or generate negatively curved deformations, which remains energetically more favorable as shown by the model presented above (Eqs. 1–9) and discuss hereafter.

**Septins avoid being positively bent**. To investigate the sensitivity of septins to a given curvature, we have designed undulated solid templates, displaying both positive (convex) and negative (concave) curvatures with tunable micrometric periodicity (see methods) (Fig. 4a).

Figure 4b shows fluorescence intensity measurements for the whole set of templates ($N_{experiments} = 15$, $N_{image\ stacks} = 202$) as a function of curvature (see methods). As expected, the density of lipids, including the density of PI(4,5)P2, remained constant within the whole curvature range. The lipid bilayer is thus homogeneous on the templates. However, the protein density varies with curvature. The density of septins is higher on positively curved than on negatively curved membranes and plateaus at $+2\,\mu m^{-1}$. This observation agrees with a previous report by Gladfelter and co-workers[21].

To understand why septins preferentially bind to positive convex curvature rather than to negative concave curvature, we used SEM (Scanning Electron Microscopy) to visualize the filaments on these patterns[29]. At low magnification, the periodic pattern was visualized by SEM and defects running perpendicular to the main waves were visible (Fig. 4c).

When bound to a flat supported membrane, septin filaments oriented randomly in parallel arrays (Supplementary Fig. 11) and their orientation seemed to be dictated by defects on the surface. However, when bound to membranes supported on wavy substrates, for a maximum and minimum curvature equal to $\pm 3\,\mu m^{-1}$ as shown in Fig. 4d–f, septins oriented differently depending on the sign of the curvature, concave versus convex

($N = 4$). On convex positive curvatures, septins lied flat along the hill axis. However, the filaments bend in the orthogonal direction along the concave negative curvature (in valleys). We quantified the relative orientation of the filaments on either concave or convex curvatures, indicated by the color code displayed in Fig. 4f, g and Supplementary Fig. 12. Filaments along surfaces at zero curvature (along wave axis) are shown in green/blue while orthogonal ones are shown in red/purple. We can pinpoint a majority of reddish filaments bending negatively within concave curvatures while filaments following the crest of convex curvatures are mostly greenish. When encountering a defect, the septin filaments change orientation to follow the direction imposed by the defect, as illustrated in Fig. 4e, g. These results clearly establish that septin organization strongly depends on substrate curvature: septins avoid positive convex curvature and align parallel to hills, whereas they have an affinity for negative concave curvature and bend negatively within valleys. We determined the relative density of filaments as a function of curvature. The relative density, i.e., (ratio of septin density measured at the point of highest positive curvature (top) versus the point of highest negative curvature (bottom)) was found to be $2.2 \pm 0.4$ ($N_{experiments} = 4$, $N_{image\ stacks} = 68$) and $2.1 \pm 0.3$ ($N_{experiments} = 2$, $N_{images} = 10$), respectively, by confocal or SEM. On positive curvatures, septins have a global tendency to bundle (see white arrow, Fig. 4f) as already shown in solution[10], which explains their higher density.

We also investigated substrates with less pronounced curvatures ($1.6\,\mu m^{-1}$ at maximum) and revealed that septins do not orient perpendicularly to the groove in these structures but adopt a tilted orientation (Supplementary Fig. 12). Moreover, after 24 h protein incubation, the global orientation of septin filaments remained unchanged, as shown in Supplementary Fig. 13. However, disruptions of the membrane visualized as cracks and holes were visible after long incubation times. In addition, occasionally, septins formed arrays of orthogonal filaments (Supplementary Fig. 14) which have been visualized in vitro using lipid monolayers[13,19] or at the bud neck of dividing *S. cerevisiae*.

These findings demonstrate that septins bend to follow negatively curved membranes while they adapt to unfavorable positive curvatures by remaining unbent at the crest of positively curved hills as displayed schematically in Fig. 4i.

To address whether the peculiar curvature sensitivity of septins at micrometer scales relies on the filamentous nature of septins or whether this is a molecular intrinsic property of the minimal nanometric septin rod, we have repeated the experiments using a mutant. We have chosen to study the behavior of a non-polymerizable mutant truncated in its alpha 0 helix[12]. This truncation precludes filament formation in the bulk. Short filaments were visualized bound to the wavy supported bilayer but they were oriented rather randomly (Fig. 4h). This assay demonstrates that pre-polymerized filaments are required to observe the curvature preference displayed in Fig. 4d–g. This was expected as the binding energy gain per octamer due to

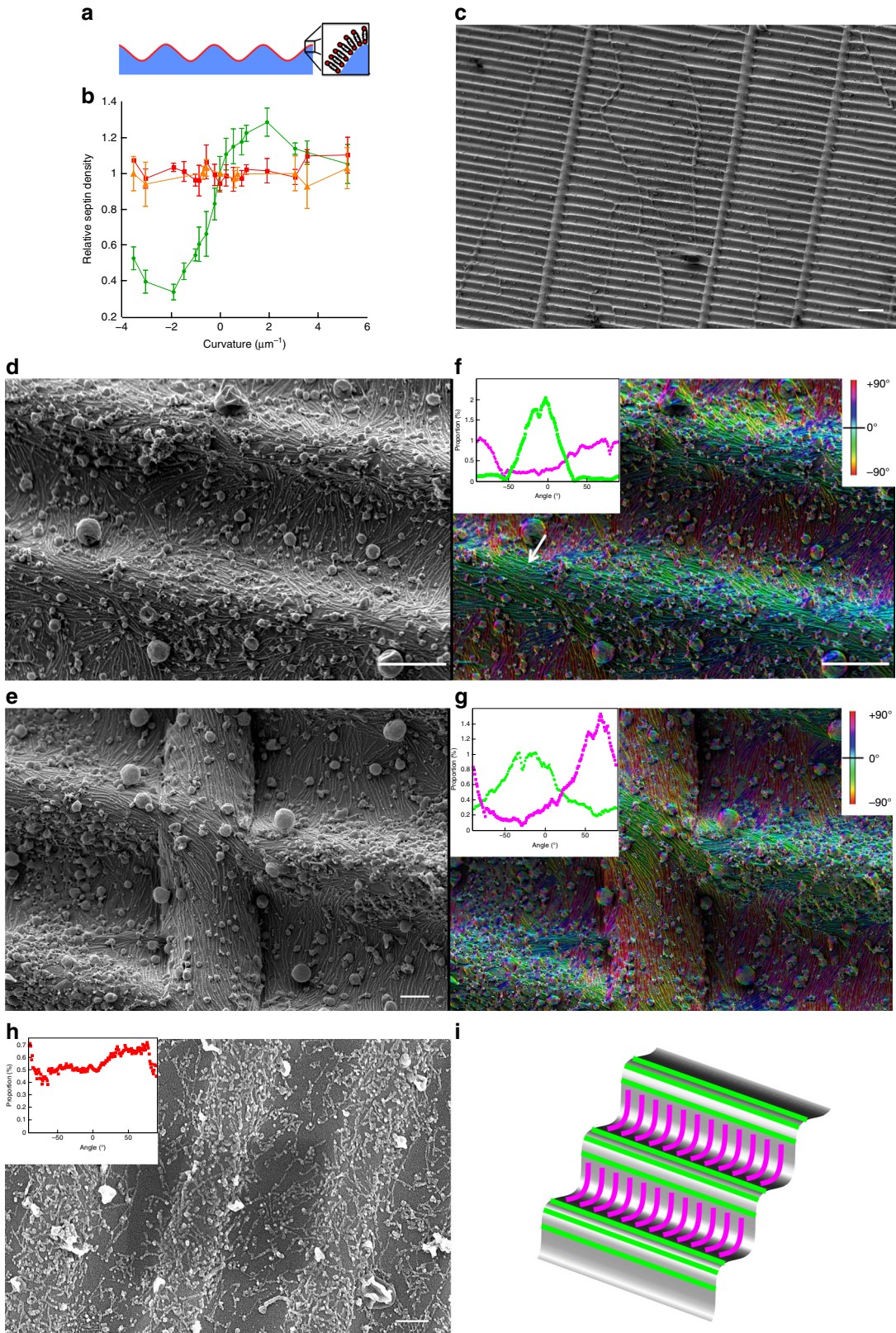

micrometric curvature, i.e., $a_1\,c\,\Delta g\,8a_1 \cong 16a_1\,c\,k_b T$, remains extremely small compared to thermal energy $k_b T$.

Using the formalism introduced above, we compared the adsorption of septin filaments on either flat or curved membranes. Equation (3) can be used to compare the adsorption of single or multiple filaments on a flat (0) versus curved (c)

membrane substrate:

$$\Delta g_n(c) - \Delta g_n(0) = -\Delta g_n^0 \frac{1}{2} a_n c + \frac{1}{2} k_b T L_{pn} c^2 \qquad (10)$$

**Fig. 4** Patterned substrates. **a** Schematic representation of the wrinkled substrates (blue) and lipid bilayer (red). **b** Relative septin density (green) on SLB deposited on the patterned substrates as a function of the curvature ($N = 202$). The red and orange curves correspond to the lipid signals as controls (Bodipy-TR-ceramide and fluorescent PI(4,5)P2, respectively). Error bars represent s.d. **c** Low magnification SEM image of a representative patterned substrate. Scale bar is 10 μm. **d**, **e** SEM image of a patterned substrate incubated with a septin bulk solution at 200 nM. Scale bar is 1 μm in **d** and 300 nm in **e**. **f**, **g** Color-labeled image of the septin orientation with respect to the wrinkle substrate of image **d** and **e**, respectively. The inset shows the angular distribution of pixels at the top (purple) (positive curvature) and at the bottom (green) (negative curvature of the substrate). The white arrow points to a convex surface where septins align and bundle parallel to the undulations. **h** SEM image of a patterned substrate incubated with mutant septin solution at 200 nM. Scale bar is 200 nm. The inset shows the angular distribution of septins. No preferential orientation is found on positive or negative curvature. **i** Schematic representation of the wavy substrates with septin filaments following the null curvature at the top, positive, (green) and the bottom, negative, curvature (purple) of the wrinkled substrate

Adsorption on curved rather than flat adsorption is favored when $\Delta g_n(c) - \Delta g_n(0) < 0$, thus, for small enough negative curvatures, $c_n < c < 0$, with

$$c_n = 2c_n^* = \frac{\Delta g_n^0 a_n}{k_b TL_{pn}} = c_1 n^{-1} \tag{11}$$

where $c_1 = \frac{1}{R_1} = 2c_1^*$, $R_1 = \frac{R_1^*}{2} \cong -0.7 \, \mu m$ and $c_2 = \frac{1}{R_2} = 2c_2^* = \frac{c_1}{2}$ with $R_2 = 2R_1 \approx -1.4 \, \mu m$. Hence, since $c_1 < c_2 < \ldots < c_n < 0$, from Eq. (11), we expect different preferential arrangements of single, double or bundle filaments depending on the local curvature $c$ with respect to $c_1, c_2, \ldots, c_n$. In particular, for $c < c_1 < c_2 < 0$ or $c > 0$, we expect only flat single and double filaments on the undulated surface, while we expect adsorption of curved single (respectively double) filaments across the groove of undulated membrane surface for $c_1^* < c < 0$ (respectively $c_1^* < c_2^* < c < 0$) (Fig. 4i). In addition, we expect possible tilted orientations for intermediary negative curvatures $c_1 < c < c_1^* < 0$ (respectively, $c_2 = c_1^* < c < c_2^* < 0$) in qualitative agreement with experimental observations (Supplementary Fig. 14).

**A theoretical model for septin remodeling in vivo.** We next considered a model to describe septin rearrangement throughout the cytokinetic process, highlighting the role of curvature and septin bundling. In vivo, septins rearrange during cell division. First, they form a collar-shaped structure where septin filaments are oriented along the bud neck axis. At the onset of cytokinesis, septins split into two rings on the sides of the contractile acto-myosin ring. Septin filaments were shown to rotate to be oriented perpendicular to the mother–bud axis[17].

We consider a 3D geometry with axial symmetry that appears more biologically relevant to septin filament throughout cellular division in vivo. We compare, in particular, radial versus circumferential adsorption of septin filaments corresponding, respectively, to parallel ($c_{\parallel}$) versus orthogonal ($c_{\perp}$) curved adsorptions with respect to the axis of symmetry of the bud.

The basic equations of filament adsorption for this cellular geometry are,

$$\Delta g_n(c_{\parallel}) = \Delta g_n^0 \left(1 - \frac{1}{2} a_n c_{\parallel}\right) + \frac{1}{2} k_b TL_{pn} c_{\parallel}^2 \tag{12}$$

$$\Delta g_n(c_{\perp}) = \Delta g_n^0 \left(1 - \frac{1}{2} a_n c_{\perp} \sin\theta\right) + \frac{1}{2} k_b TL_{pn} c_{\perp}^2 \tag{13}$$

where $\theta$ is the angle of the circumferential filament plane with the membrane. We consider in more details the two cases, $\sin\theta = \frac{R_{\perp}}{R_{\parallel}}$ (spherical shape, Fig. 5a) and $\sin\theta = 1$ (oblong shape, Fig. 5b), below.

First, for a spherical cell of radius $R_{\parallel}$, we have $\sin\theta = \frac{R_{\perp}}{R_{\parallel}}$ and thus $c_{\perp}\sin\theta = c_{\parallel}$, which cancels the difference in area change contribution between radial and circumferential filaments. Hence,

comparing the two orientations of adsorption, we obtain

$$\Delta g_n(c_{\parallel}) - \Delta g_n(c_{\perp}) = \frac{1}{2} k_b TL_{pn}(c_{\parallel}^2 - c_{\perp}^2) \tag{14}$$

which implies that single or multiple filaments will always adopt a radial orientation, as $|c_{\parallel}| < |c_{\perp}|$, i.e., $|R_{\parallel}| > |R_{\perp}|$, in this global (or local) spherical geometry (Fig. 5a). This situation is consistent with the early organization of the septin ring before the bud is formed[15].

We then consider a general oblong shape of the (dividing) cell assuming only axial symmetry. We approximate Eqs. (12) and (13) by considering $\theta \approx \frac{\pi}{2}$ and thus $\sin\theta \approx 1$ for these oblong shapes, yielding

$$\Delta g_n(c_{\parallel}) = \Delta g_n^0 \left(1 - \frac{1}{2} a_n c_{\parallel}\right) + \frac{1}{2} k_b TL_{pn} c_{\parallel}^2 \tag{15}$$

$$\Delta g_n(c_{\perp}) \approx \Delta g_n^0 \left(1 - \frac{1}{2} a_n c_{\perp}\right) + \frac{1}{2} k_b TL_{pn} c_{\perp}^2 \tag{16}$$

This enable us to compute a simple condition for parallel versus perpendicular septin filament adsorption in these oblong cellular geometries with an axial symmetry, corresponding to $c_{\parallel} - c_{\perp} > 0$,

$$\Delta g_n(c_{\parallel}) - \Delta g_n(c_{\perp}) = -\frac{1}{2} \Delta g_n^0 a_n(c_{\parallel} - c_{\perp}) + \frac{1}{2} k_b TL_{pn}(c_{\parallel}^2 - c_{\perp}^2) \tag{17}$$

Leading to a transition of septin filament adsorption from the parallel (radial) to perpendicular (circumferential) arrangements when $c_{\parallel}$ becomes smaller (cylinder shape) and ultimately negative, corresponding to a membrane constriction between the mother and daughter cells (Fig. 5b–e). This transition from parallel (radial) to perpendicular (circumferential) adsorptions of septin filaments corresponds to,

$$0 < (c_{\perp} + c_{\parallel} - c_n)(c_{\parallel} - c_{\perp}) \tag{18}$$

$$0 < \left(\frac{1}{R_{\perp}} + \frac{1}{R_{\parallel}} - \frac{1}{nR_1}\right)\left(\frac{1}{R_{\parallel}} - \frac{1}{R_{\perp}}\right) \tag{19}$$

where $c_n$ is given by Eq (11) $c_n = \frac{\Delta g_n^0 a_n}{k_b TL_{pn}} = c_1 n^{-1}$, where $R_1 = -0.7 \, \mu m$.

Hence, since $c_1 < c_2 < \ldots < c_n < 0$ from Eq. (11), we expect different preferential arrangements of single, double or bundle filaments depending on the local curvature $c_{\parallel} + c_{\perp}$ with respect to $c_1, c_2, \ldots, c_n$. In particular, for an oblong cell with axial symmetry, corresponding to $|c_{\parallel}| < |c_{\perp}|$ (Fig. 5b–e), the model predicts that single filaments adopt a circumferential adsorption while double

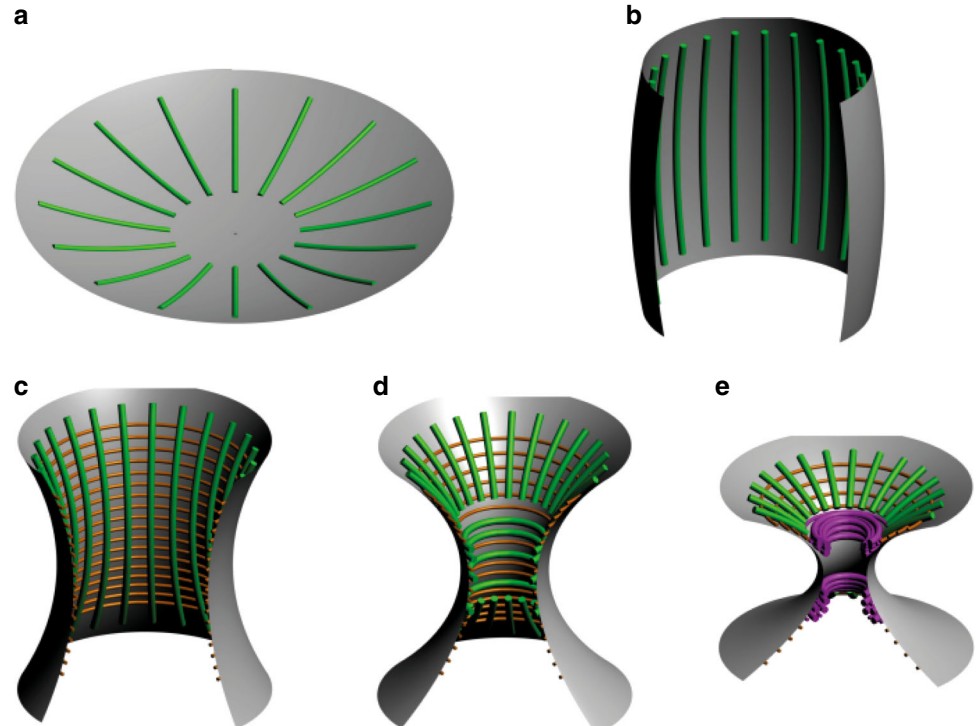

**Fig. 5** Schematic representation of model predictions for septin filaments at the bud neck of *S. cerevisiae*. **a** Filaments on a spherical geometry arranged radially. **b** Filaments on a barrel-like geometry arranged along the height of the barrel. **c** At the onset of the constriction, single filaments are arranged circumferentially while double filaments are arranged perpendicular to the single filaments. **d** Upon further constriction, all filaments at the center of the bottleneck are arranged circumferentially while the organization remains the same at the edges. **e** Under extreme constriction, filament bundles appear

filaments are still expected to remain parallel to the cylindrical symmetry axis for $c_1 < c_\perp + c_\parallel < c_2 = \frac{c_1}{2} < 0$ (Fig. 5c).

Then, upon further membrane constriction between the mother and daughter cells, $c_1 < c_2 < c_\perp + c_\parallel$, both types of single and double septin filaments are predicted to adopt a circumferential geometry close to the saddle point of the membrane neck between the two cells (Fig. 5d).

Finally, under the condition that single and double filaments can form thicker filament bundles in the neck region (which is known to require additional molecular players such as *myo1* proteins and *actin* proteins[30]), we expect that these bundles will eventually reach their minimum radius of curvature under membrane adsorption, given by Eq. (22) (see below). Indeed, Eq. (3) predicts a maximum (positive or negative) curvature of adsorbed septin filaments on a curved membrane substrate. Its magnitude can be estimated by neglecting the change of contact area due to curvature ($ca_n \ll 1$) in Eq. (3) as,

$$\Delta g_n(c) = \frac{1}{2} k_\mathrm{b} TL_{pn} c^2 + \Delta g_n^0 \qquad (20)$$

Which leads to a maximum curvature $c_n^\mathrm{max}$ and minimum radius of curvature $R_n^\mathrm{min} = \frac{1}{c_n^\mathrm{max}}$ of multiple septin filaments adsorbed on a curved membrane,

$$c_n^\mathrm{max} = \sqrt{\frac{-2\Delta g_n^0}{k_\mathrm{b} TL_{pn}}} = c_1^\mathrm{max} n^{-3/4} \qquad (21)$$

$$R_n^\mathrm{mix} = \sqrt{\frac{k_\mathrm{b} TL_{pn}}{-2\Delta g_n^0}} = R_1^\mathrm{min} n^{3/4} \qquad (22)$$

where $R_1^\mathrm{min} = \sqrt{\frac{kTL_{p1}}{-2\Delta g_1^0}} \approx 37$ nm. Hence, the minimum radius of curvature depends on the number $n$ of filaments in the septin multiple filament bundle as, $R_2^\mathrm{min} \approx 63$ nm, $R_{10}^\mathrm{min} \approx 208$ nm, $R_{30}^\mathrm{min} \approx 474$ nm, and $R_{50}^\mathrm{min} \approx 696$ nm.

Ultimately, this minimum radius of curvature constraint might drive the separation of the septin assembly at the neck into characteristic double septin rings including several tens of filaments on each side of a thinner membrane bridge across the two cells (Fig. 5e). Typical radii of curvature of the double septin rings[14,31] are comparable with the minimum radius of curvature estimated here from Eq. (22). i.e., 0.5–1 μm, although other molecular interactions might also contribute to the maximum curvature of septin filaments.

## Discussion
Our observations demonstrate that the organization of septins depends on curvature and that the deformations of both GUVs and LUVs originate from this specificity.

While septins do not exhibit curvature sensitivity per se, and can bind to both positively and negatively curved membranes, we find that they organize differently on positively or negatively curved templates. On negatively curved substrates septins, bound to lipids, have the ability to adopt a negative curvature, whereas on positive geometries septins align along the axis of null curvature. Parallel to one another and unbent on a surface, the filaments have a tendency to bundle. These observations explain the higher septin density measured on positive geometries. This pattern resembles the organization of septins observed within spheroplasts[15]. Using silica beads, the Gladfelter lab has shown that septins bind preferentially to beads of 2 μm⁻¹ curvature[22]. Using an appropriate lipid composition, septins certainly have the

ability to interact with membrane-coated silica beads being filamentous or rod-like.

Indeed in our LUVs assays, septins do interact with vesicles displaying a diameter above 100 nm. Highly curved vesicles (LUVs) are flattened by septin filaments lying flat parallel to one another. However, GUVs are more drastically deformed by septin filaments. Spikes are induced with a regular micrometric spacing, and are connected by negatively curved membranes. Connecting each spike, septin filaments are most likely negatively bent, resembling their organization within the negative curves of undulated substrates. Bound to spikes and small protrusions, septin filaments, as shown by cryo-EM, organize parallel to the axis of the protrusion. Our report thereby demonstrates that septin filaments seek alternative strategies to avoid being positively bent by deforming their positively curved substrate.

In general, curvature sensor proteins have been reported to sense nanometric curvatures[19,32]. The intrinsic shapes of those proteins, such as the crescent-like shape of BAR proteins, for instance, and their strong interaction with membranes dictate their ability to sense curvatures at the molecular scale. Only a restricted number of proteins are known to interact specifically with micrometric curvatures[33]. Septins[22], FtsZ[34], MREB[35], and SpoVM[36] have been shown to interact on micrometric curvatures with an enhanced affinity. An amphipathic helix within SpoVM has been shown to be responsible for the interaction[36]. Budding yeast–septins interact with the membranes through basic stretches of amino acids[12]. The septin membrane interaction is not ruled at the subunit or octameric scale. We have indeed shown that the observed behavior originates from the filamentous nature of the septins. Using a septin mutant that does not polymerize in bulk, we do not recover any specific curvature sensitivity on the undulated patterns. Besides, the proposed theoretical model to account for the organization of septins and membrane deformations considers the septins as pre-formed filaments. We note that, similar deformations of GUVs induced by other filamentous proteins (FtsZ) are observed[34].

Not only does the minimal theoretical model account for our experimental in vitro observations but it also gives insights on the remodeling of the septin filaments throughout cytokinesis. Indeed our model explains how the filaments might adjust from an arrangement parallel to the bud axis to a circumferential arrangement during cytokinesis. Besides, we infer that the bundling of the filaments, might favor their splitting into two individual bundled rings due to the maximum curvature constraint at the bud neck. Hence, it shows that the geometry and curvature in situ have a role in tuning the organization of septins in concert with post-translational modifications. In bacteria, the orientation of MreB filaments is also correlated with the bacteria cell shape[35].

We may wonder whether septins are also involved in membrane deformations in cellulo. Septins are found at sites where membrane is strongly remodeled. While actin and myosins are believed to have a major mechanical active role in these processes, septins have been considered as a passive scaffold to recruit essential components. However, our results suggest that the role of septins might have been underestimated throughout those processes. Our biophysical findings might also apply or have consequences for higher eukaryotes as well. For instance, Tanaka-Takiguchi et al.[37] have shown that GUVs are spontaneously remodeled by human septins (Sept2-Sept6-Sept7), reporting protrusions and tubular structures. There, human septin rings decorate tubules in a concentric manner while in our study, budding yeast–septins assemble parallel to the protrusions. These differences might reflect specific septin functions which might be species-dependent and tissue-dependent.

Our findings demonstrate that the filamentous nature of septins dictate specific arrangements on curved substrates and thus membrane deformations. Besides, using a simple theoretical model, we suggest that curvature has a role in the rearrangement of Septin filaments during cell division.

## Methods

**Reagents**. Common reagents (ethanol, acetone, chloroform, sucrose, sodium chloride, Tris) were purchased from VWR reagents. L-α-phosphatidylcholine (EPC, 840051P), cholesterol (700000P), 1,2-dioleoyl-sn-glycero-3-phosphoethanolamine (DOPE, 850725P), 1,2-dioleoyl-sn-glycero-3-phospho-L-serine (DOPS, 840035P), and L-α-phosphatidylinositol-4,5-bisphosphate (PI(4,5)P2 840046P) were purchased from avanti polar. Bodipy-TR-ceramide was purchased from invitrogen (D-7540).

**Protein purification**. Yeast–septins complexes containing Cdc11, His6-Cdc12, GFP-Cdc10, and Cdc3 were co-expressed in *Escherichia coli* and purified as described in detail elsewhere[10]. Briefly, septins are purified by immobilized nickel affinity, size exclusion and ion exchange chromatography. A Cdc11_His6-Cdc12_Cdc3_GFP-Cdc10_GFP-Cdc10_Cdc3-His6-Cdc12_Cdc11 fluorescent and palindromic complex is thus obtained.

**Preparation of giant unilamellar vesicles**. Giant unilamellar vesicles (GUVs) were formed using electro-formation on platinum wires as previously described[26]. Vesicles were formed from a mix containing EPC, 10% DOPS, 10% DOPE, 15% cholesterol, 0.5% Bodipy-TR-ceramide, and various amounts of PI(4,5)P2 lipids up to 10% (molar fractions). 4–5 μL of the lipid mixture at 3 mg mL$^{-1}$ were spotted and dried under vacuum for 30 min, on two platinum wires mounted in a custom-made teflon chamber. The lipids were rehydrated in 10 mM Tris pH 7.6, 50 mM NaCl, 50 mM sucrose, under an AC voltage of 350 mV at 500 Hz for 6 h at +4 °C. GUV were then collected and transferred in an iso-osmotic observation buffer (10 mM Tris pH 7.6, 75 mM NaCl). In some cases, a higher salt concentration was used in the observation chamber (NaCl 300 mM instead of 75 mM), the osmolarity was adjusted accordingly by adding sucrose in the growth buffer.

**Pipette experiment to determine the bending modulus**. Micropipette aspiration was used to determine the bending modulus of vesicles (as first described by Kwok and Evans[38]), GUVs were inserted into an home-made observation chamber mounted on an inverted microscope (Eclipse TE2000 Nikon (Japan), ×60 water objective). The setup chamber was equipped with two micropipette holders connected to a water reservoir to control the pressure: one pipette holding the GUV and the other one injecting proteins. GUVs were held using the suction micropipette ($R \approx 3$ μm) thereby creating a tongue. The water reservoir allows applying a pressure difference between the interior of the GUV and the pipette ranging from 10 to 1000 Pa. The membrane tension σ of the GUV is controlled by the applied pressure difference and vary from $10^{-6}$ to $10^{-3}$ N m$^{-1}$.

At low tensions (σ < 0.5 mN m$^{-1}$ that corresponds to P < 350 Pa), thermal fluctuations persist and can be assessed by applying step-by-step tension increments. At each tension step, an image was collected by confocal microscopy after 60 s to ensure that equilibrium was reached. The bending modulus can be extracted by looking at the variation in the projected area in response to changes in membrane tension between a reference $\sigma_0$ and **σ**, following the equation[25]:

$$\ln\left(\frac{\sigma}{\sigma_0}\right) \approx \frac{8\pi \cdot \kappa}{k_b T} \frac{\Delta A}{A_0} \qquad (23)$$

where $\kappa$ is the bending modulus, $k_b T$ is the thermal energy, $\Delta A$ is the change in projected area in response to a change of tension and $A_0$ is the initial area of the GUV.

The experiments were performed using confocal fluorescence imaging. Supplementary Fig. 1 shows plots of representative measurements of the bending modulus of single GUVs with or without septins.

**Confocal fluorescence imaging and image analysis**. Confocal experiments were performed on a Nikon Eclipse TE2000 inverted microscope. The software EZ-C1 was used to make the acquisition of the confocal images. The imageJ radial profile plugin was used to analyze the fluorescence signal on the GUVs. Protein density was measured as described in supplementary material of Aimon et al.[39] Briefly, the fluorescence signal of a reference fluorescent lipid (Oregon-Green® 488 DHPE) was measured on GUVs at known densities. The fluorescence of GFP-septins and Oregon-Green was subsequently compared in solution. Knowing the geometrical parameters of septin filaments and the relative fluorescence intensity of GFP and the reference fluorophore, we were able to deduce the density of septins bound to a GUV from the intensity of fluorescence signal.

**Spinning disk microscopy and 3D reconstitutions**. Spinning disk microscopy was performed using an inverted spinning disk confocal Roper/Nikon equipped

with a CSU-X1 Yokogawa head, a ×100 oil objective and two laser lines (491 and 561 nm). Image stacks resulted from collecting images at 0.4 μm spatial intervals. 3D color reconstitutions were obtained using ImageJ coloring and 3D projection tools.

**Sample preparation and imaging for cryo-electron microscopy**. A lipid mixture (70% EPC, 10% DOPE, 10% DOPS, 10% PIP2) at 1 mg mL$^{-1}$ was quickly dried under argon for 2 min then put under vacuum for 30 min. LUVs of variable size (50–500 nm) were obtained by resuspension and vortexing of the lipid film after addition of a buffered solution to reach a final concentration of 0.1 mg mL$^{-1}$. Thanks to the presence of anionic lipids in the solution, the proportion of multilamellar vesicles remains low. Septins were incubated with the vesicles at a lipid:protein weight ratio of 5:1 for about 1 h. A 5 μL drop of the solution was deposited on a glow discharged lacey carbon electron microscopy grid (Ted Pella, USA). Most of the solution was blotted away from the grid to leave a thin (<100 nm) film of aqueous solution. The blotting was carried out on the opposite side from the liquid drop and plunge-frozen in liquid ethane at −181 °C using an automated freeze plunging apparatus (EMGP, Leica, Germany). The samples were kept in liquid nitrogen and imaged using a Tecnai G2 (FEI, Eindhoven, Netherlands) microscope operated at 200 kV and equipped with a 4kX4k CMOS camera (F416, TVIPS). The imaging was performed at a magnification of 50,000 with a pixel size of 2.13 Å using a total dose of 10 electrons per Å$^2$.

**Cryo-tomography**. The samples were prepared as described above. 10 nm size gold beads were added to the solution before being plunge-frozen. Tilt series were collected in low dose mode, every two degrees, using a Tecnai G2 (FEI, Eindhoven, Netherlands) microscope operated at 200 kV and equipped with a 4kX4k CMOS camera (F416, TVIPS). Images were collected following this angular scheme: 0 to −34°, then +2 to 60° and finally −36 to −60° to minimize irradiation at the lowest angles. The dose per image was 0.8 electrons per Å$^2$. The imaging was performed at a magnification of 50,000 and each image was binned twice for a final pixel size of 4.26 Å. The consecutive images were aligned using the IMOD software suite[40]. Back projection was performed using IMOD and SIRT reconstruction was carried out using Tomo3d[41]. The segmentation was performed manually using IMOD.

**SUVs formation to assemble supported bilayers**. Lipid mixtures composed of (EPC 56.5%, Cholesterol 15%, PI(4,5)P2 8%, DOPE 10%, DOPS 10%, Bodipy-TR-ceramide 0.5%; % are in molar fractions) were mixed in a glass vial and dried under a nitrogen flux for 5 min and further dried under vacuum for 30 min. The lipids were then rehydrated at 4 mg mL$^{-1}$ in 20 mM citric acid pH 4.8 + 150 mM KCl for 30 min. The mixture was vortexed for a few seconds and sonicated for at least 15 min in a water bath sonicator until a transparent solution was obtained. Aliquots were stored in a freezer at −20 °C for up to 2 weeks for later uses.

Solution of SUV were thawed and diluted at 1 mg mL$^{-1}$ in 20 mM citric acid at pH 4.8 150 mM KCl and 5 mM CaCl$_2$. 75 μL of the solution was deposited on the NOA wavy replicate, previously treated with plasma for 2 min to create a supported bilayer. After 1 h, the substrates were washed thoroughly using the observation buffer (75 mM NaCl, 10 mM Tris pH 7.6). After washing septins were deposited on the substrate in the observation buffer at concentrations ranging from 100 to 600 nM and incubated for at least 20 min. The septins in excess are then washed off. Sample were then observed using confocal fluorescent microscopy or further treated with chemical fixation to performed scanning electron microscopy.

**Design of undulated patterns**. Wrinkled poly(dimethyl siloxane), PDMS, stamp substrates were designed and fabricated as described by Nania et al.[42] by plasma exposure and uniaxial compression. Surface replicates were obtained by depositing the PDMS stamp on top of a small amount (~2 μL) of a thiol-ene based UV-curable adhesive (Norland Optical Adhesive NOA81) placed on a glass coverslip, previously washed in ethanol and dried. The NOA-coated glass coverslip was then placed under a UV light for 10 min to polymerize the adhesive. The PDMS stamp was then carefully peeled from the coverslip and stored for further replication. The NOA wavy substrate can then be stored or used immediately.

We vary the amplitude of the templates from 0.6 μm up to 1.1 μm and the longitudinal periodicity from 1.8 to 5.8 μm. We have selected patterns with curvatures varying from -3.6 μm$^{-1}$ to +5.2 μm$^{-1}$. Lipid bilayers containing PI(4,5)P2 and red fluorescent lipids were deposited onto the substrates before incubation with GFP-tagged septins in polymerizing conditions[43].

Using confocal fluorescence microscopy, we measured the density of both lipids and septins as a function of curvature as explained in the material and methods section. The fluorescence signals of lipids and septins were measured by vertically scanning surface sections every 300 nm, corresponding to specific curvatures.

**Sample preparation and imaging for SEM**. NOA wavy substrate covered with lipids and incubated with 100 nM of septins were fixed with Glutaraldehyde 2% in sodium cacodylate 0.1 M for 15 min. The samples were washed three times with sodium cacodylate 0.1 M and incubated for 10 min with a second fixative (Osmium tetroxide 1% in sodium cacodylate 0.1 M). After three washes with water, the samples were incubated with tannic acid 1% for 10 min and subsequently washed

three time with water before being incubated with Uranyl acetate (1% in water) for 10 min. The samples were then dehydrated using baths with increasing ethanol concentrations (50, 70, 95, 100%) and processed using a critical point dryer (CPD 300, LEICA). After being mounted on a sample holder, the samples were coated with 1.1 nm of either tungsten or platinum (ACE 200, LEICA). SEM imaging was performed using a GeminiSEM 500 microscope from Zeiss, Germany.

SEM images were analyzed using OrientationJ (a plug-in for imageJ) to determine the orientation of the filaments (http://bigwww.epfl.ch/demo/orientation/). After sharpening, the filaments are detected and the orientations of the filaments are displayed using a color code. To extract quantitative information, the signal coming from the defects (small vesicles) was highlighted and subtracted. Graphs were plotted to display the distribution of the filaments depending on their orientation, with 0° corresponding to the orientation along the waves. The green curves describe the distribution measured on positive curvatures (hills) and the purple/red ones on negative curvatures (valleys) (see inserts).

**Determination of $K_d$ dissociation constant**. SLBs were prepared as described above, on flat NOA surface. Septins were then added at final concentration up to 250 nM. Confocal images were taken 15 min after septins were added. The septin density on the surface was determined and plotted for each septin bulk concentration. Septin density was measured using the protocol of Aimon et al.[39] explained in the "Confocal fluorescence imaging and image analysis" section above. Curves were fitted by the following Hill equation

$$[\text{septin}]_{\text{bound}} = \frac{S_{\text{sat}}}{\left(\left(\frac{K_d}{[\text{septin}]_{\text{bulk}}}\right)^n + 1\right)} \tag{24}$$

where [septin]$_{\text{bound}}$ is the measured septin density bound to the SLBs, $S_{\text{sat}}$ is the saturation density, $K_d$ is the dissociation constant, [septin]$_{\text{bulk}}$ the septin bulk concentration, and $n$ the Hill parameter. Supplementary Fig. 4 shows the result of these measurements.

**Persistence length measurement**. Budding yeast–septins were diluted at 5 nM into a low salt buffer including methylcellulose (20 mM imidazole-HCl at pH 7.4, 1 mM dithiothreitol, 0.1 mM MgATP, 50 mM KCl and 2 mM MgCl$_2$, 1 mM Trolox, 2 mM protocatechuic acid, 0.1 μM protocatechuate 3,4-dioxygenase, and 0.1% (w/v) methylcellulose) and inserted into a imaging chamber which surface had been passivated for 45 min using KOH 1 M followed by 45 min PLL-PEG 0.2 mg mL$^{-1}$.

To measure the persistence length of septin filaments, we studied their fluctuations on the passivated surface by TIRF microscopy. To obtain a measurement of the persistence length of a fluctuating filament, we considered a vector $\overrightarrow{A_0}$ at position 0, tangent to the filament and a vector $\overrightarrow{A_L}$ tangent to the filament positioned at an arc length $L$ of $\overrightarrow{A_0}$. Noting $\theta$ the angle between those two vectors, the orientation persistence along the filament can be described as:

$$\langle \cos \theta \rangle = e^{\frac{(d-1)L}{2L_p}} \tag{25}$$

where the brackets denote the average over all starting position 0, $d$ is the dimension (here $d = 2$) and $L_p$ is the persistence length of the filament.

The filaments were tracked using the JFilament plug-in of imageJ (https://imagej.net/JFilament). Supplementary Fig. 6A shows a raw image of a septin filament and the same filament tracked by the software. This gives 2D coordinates for all points along the tracked filaments. With a matlab script, we extracted from these coordinates the tangent vectors and calculated $\langle \cos\theta \rangle$ for each value of $L$. We then plotted $\langle \cos\theta \rangle$ against $L$ in a semi-log scale and extracted the persistence length value from the slope. Supplementary Fig. 6B displays the final curve, resulting from an averaging over 300 images corresponding to 12 different septin filaments.

## Data availability

Data supporting the findings of this manuscript are available from the corresponding authors upon reasonable request. Data that support the findings of this study have been deposited in EMBD with the EMD-0321 accession code.

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

## Acknowledgements

This work was supported by Institut Curie, Centre National de la Recherche Scientifique (CNRS), the Agence Nationale pour la Recherche (grant ANR-13-JSV8-0002-01 to A. Bertin). We thank the Cell and Tissue Imaging (PICT-IBiSA), Institut Curie, member of the French National Research Infrastructure France-BioImaging (ANR10-INBS-04). We acknowledge Gérard Péhau-Arnaudet at the Imagopole (Institut Pasteur) for some cryo-EM imaging. We thank Virginie Bazin and Mickaël Trichet from the Electron Microscopy facility of the Institut de Biologie Paris-Seine (IBPS, Paris) for some of the SEM sample preparation and technical assistance during acquisitions. We thank Manos Mavrakis and Gijsje Koenderink for sharing the TIRF data on passivated surfaces. Alexandre Beber was funded by Ecole doctorale PIF, Physique Ile de France. D.L. and P. B. groups belong to Labex CelTisPhyBio (ANR-11-LABX0038) and to Paris Sciences et Lettres (ANR-10-IDEX-0001-02).

## Author contributions

A.Bertin, S.M. and A.Beber designed research. D.L. and P.B. provided conceptual advice. A.Bertin expressed and purified the septin complexes. A.Bertin, C.T., D.L. and A.D.C. performed the cryo-electron microscopy experiments. M.N. and J.C. designed the PDMS substrates for the SEM experiments. A.Bertin and A.Beber carried out the SEM experiments. S.M. and A.Beber performed the fluorescence microscopy experiments. H.I. generated the theoretical model. F.C.T. carried out the TIRF experiments on passivated surfaces. The manuscript was written by A.Beber, A.Bertin, S.M. and H.I. The results and their interpretation were discussed by all of the authors.
