## [Peer Review File · Nature Communications]

Reviewers' Comments:

Reviewer #1:

Remarks to the Author:

Manuscript ID: NCOMMS-18-20375

Manuscript entitled "Membrane reshaping by micrometric curvature sensitive septin filaments" by Alexandre Beber et al. investigated the relationship between the membrane morphogenesis done by septin filament and its curvature sensitivity using supported lipid bilayers that were prepared on custom designed periodic wavy patterns displaying positive and negative micrometric radii of curvatures, and demonstrated that the organization of septins is sensitive to curvature and that the deformations of lipid bilayer membranes, as well as cellular membranes, originate from this specificity.

Both the method used and results obtained in this study are very interesting. Thus, I would like to suggest that the manuscript will be published in Nature Communications.

Some comments and questions are below:

(1) Lines 133 – 134 of page 6: Those bundles were most likely self-assembled in solution **prior** to their interaction with the vesicles.

Why can you consider so? What is the specific evidence?

(2) Lines 193 and 195 of page 8 (Eqs. 5 and 7):

Please unify the indication style of superscript, multiplier, of n.

(3) Line 240 of page 9:

... (Figure 3 D, left panel, **arrow**).

In the left panel of Figure 3D, the arrow is missing.

(4) Lines 242 - 243 of page 9:

Overall, these observations demonstrate that septins deform and constrain small positively curved vesicles (diameters from 100 nm to 1 μ m) into **flat "pancakes"**, ...

It is difficult for me to understand the result indicating that small positively curved vesicles deformed into the flat pancakes. If I look at which part of which Figure, can I understand it?

I hope a dedicated Figure to show that morphology.

(5) Lines 259 -261 of page 10:

Figure 4.B shows fluorescence intensity measurements for the whole set of templates (Nexperiments = 15, Nimage stacks = 202). As expected, the density of lipids remained constant within the whole curvature range. The lipid bilayer is thus **homogeneous on the templates**.

I agree about the density. But can lipid composition be uniform as well?

Especially, since the head group of PIP2 is very large, is there a possibility that the content will be locally different depending on the local curvature?

(6) In Figure 4F (page 23):

Although there is one white arrow, the explanation about it is missing.

(7) The legend of Figure 1C-E (page 25):

Do the error bars indicate S.D.? (or S.E.?)

(8) Lines 568 – 569 in the legend of Figure 4F-G (page 26):

The inset shows the angular distribution of pixels at the top (**red**) (positive curvature) and at the bottom (**black**) (negative curvature of the substrate).

Are they purple and green, respectively?

(9) On the Methods, no reference is cited in some sections about cryo-EM, SUV formation to assemble supported bilayers, SEM, and persist length measurement.
Are there no cited references to serve as reference?

(10) Line 157 of the "Persist length measurement" section of the Methods:
The composition of the solution used in the experiment is much different from the others.
Especially, it contained divalent cation and methylcellulose.
Is there no problem?

(11) Lines 169 and 174 of the "Persist length measurement" section of the Methods:
Fig S5.A and Fig S5.B should be Fig S6.A and Fig S6.B, respectively.

(12) About Supplementary Figure 1:
It is divided into two panels (cases without and with septin).
But, at least to me, it is easier to understand, when it is put together into one panel.

(13) As the authors mentioned, septins are guanine-nucleotide-binding proteins.
However, there is no description about the addition of GTP anywhere, and none of the solutions contain GTP.
Is there no problem?

Reviewer #2:

Remarks to the Author:

I am serving as a technical reviewer for the cryo-electron tomography (cryo-ET) experiments presented in this paper.

The authors use a number of biophysical methods including cryo-ET to describe how septins interact with differentially curved membrane bilayers in vitro. These results are used to develop a model for how septins might promote cell division.

As I am not an expert on septin biology I cannot judge the novelty and the impact of the paper on the field.

It is good practice in the cryo-EM field to deposit reported data for public access after publication. The authors should submit at least one example tomogram in the Electron Microscopy Data Bank (EMDB) and include the accession code in the paper.

There are several aspects regarding the interpretation of the cryo-ET experiments that should be addressed in a revised version of the manuscript.

1) Major concerns:

- Visual/manual filament segmentation is problematic, because visibility of filamentous structures to the human eye in cryo-ET data strongly depends on their orientation with respect to the imaging direction and the tilt axis. In particular, filaments oriented perpendicular to the imaging direction (i.e. in the sample plane) can be typically recognized and traced much better than filaments oriented parallel to the imaging direction. Could the authors try and trace/segment the septin filaments in an automatic fashion to support their manual segmentation?
- The authors suggest that septin filaments deform vesicles into flattened "pancakes". This is a very problematic conclusion, because isolated vesicular structure always appear significantly flattened in cryo-ET data, most likely because they are squeezed by the surface tension of the water/buffer film during the blotting process that is part of the cryo-EM grid preparation. There are numerous published cryo-ET studies that document this effect. The authors must consequently test and document systematically that the flattening of vesicles they observe in their data indeed

also appears in directions not expected due to the effects of the surface tension.

2) Minor points:

2.1) Abstract:

- line 15 and several additional instances: "inner plasma membrane" should probably be "inner face of the plasma membrane", or "cytoplasmic face of the plasma membrane"
- "septin filament curvature arrangement preferences" is very complicated. Please provide more comprehensive description.

2.2) Introduction:

- I think there are several aspects in the paper that could benefit greatly from visualization in a Figure:
 - * basic palindromic layout of septin filaments (line 40)
 - * architecture of minimal septin oligomers (line 42)
 - * geometry of bud necks in dividing cells and septin filament arrangement described in the third introduction paragraph (lines 46-53)
- line 61: "proteins can also form a coat interacting with the membrane TO induce membrane deformation"

2.3) Results:

- line 93: what is the physiological septin concentration? Do the concentrations used in the experiments recapitulate the physiological concentration range?
- line 104: "septins. μm^{-2} " should be "septins/ μm^{-2} "
- line 146-148: has it been shown previously that electrostatic interactions do not play a role anymore at $> 150 \text{ mM NaCl}$?
- Fig. 3A: please enlarge the inserts. They are almost impossible to recognize in a printed version.

2.4) Methods:

- line 89: please add the total electron dose of 2D projections.
- line 93: "tilted series" should be "tilt series"
- line 93: please specify the tilt scheme: were the tilt series acquired in one go from -60° to 60° , was it a bi-directional scheme starting at some low tilt angle, or were the authors using the dose-symmetric "Hagen scheme"?
- line 95: the authors claim that they acquired tilt images with a constant dose of $0.8 \text{ e}^-/\text{Å}^2$ for each tilt image. This is rather unusual, because exposure is normally adjusted to the increasing sample thickness upon tilting, e.g. in a cosine like manner.
- line 95: what is the cumulative dose used for acquisition of the entire tilt series?
- line 231: please describe how the statistical analysis of filament arrangement based on the cryo-ET data was done.
- line 234: please visualize schematically the arrangement described here
- line 240: the authors observe filaments rupturing the vesicles. Do they expect this to happen in vivo? If so, what would be the implications?

Reviewer #3:

Remarks to the Author:

Septins are cytoskeletal filaments that assemble at the inner plasma membrane, localize at constriction sites during cytokinesis and impact membrane remodeling processes. We report a range of in vitro biomimetic tools to examine how yeast septins behave on curved and deformable membranes.

The authors use a combination of membrane biophysical tools, microscopy, and cryo EM coupled with simple mathematical modeling to investigate how septin assembly couples to membrane curvature.

The curvature sensitivity was established in GUV as well as in wavy substrate experiments and the septin assembly was investigated through EM studies.

The basic premise is that the adhesion of septin to membrane does not change the mechanical properties of the membrane and so the adhesion strength directly trades with membrane curvature energy. Assuming adhesion per unit length does not change on septin deformation, one can develop a conceptual model for total energy minimization and predict the minimum energy of septin assembly in different orientations in planar versus curved interfaces.

The experiments are done cleanly and the results are described clearly however, the methods and the results provide plausible explanation of the experimental results and does not probe the results or the mechanism deeply.

Here are some points of concern:

1. Polymerization free energy of septin can be curvature dependent. Energy of bending septins can also penalize the total energy. Both factors are not accounted for in the model
2. Are entropic terms -- conformational entropy of septins or the membrane important? How can the authors rule this out?
3. Does PIP2 distribution on the membrane depend on curvature? The current model assumes it is not. If so the curvature dependence of septin adhesion can be explained using PIP2 spatial distribution coupled to curvature.
4. It would be more compelling if the authors performed experiments at different PIP2 concentrations and also develop a model that can account for this PIP2 dependence.
5. The authors can also try and manipulate the spatial organization of PIP2 -- say by cholesterol or other means and challenge the mathematical model
6. It is not clear why membrane tension does not contribute to the explanation. Can the authors perform the experiments under different tension? Can the mathematical model be formulated to include tension? Can a comparison between the model and experiment be made in a tension dependent fashion?
7. I think by adding extra dimensions to challenge the model against experiment (PIP2, tension) is important to establish unequivocally the explanation for the results. Otherwise the authors are just providing a plausible explanation and not a definitive one.
8. Does inter filament interaction contribute to the assembly? Can the authors comment on how or manipulate this?

I am concerned that while the observations are very interesting they are not developed enough for this study to be authoritative.

I am sorry I am unable to be more encouraging, but I do recognize at the same token that the results are promising.

We thank the referees for their relevant comments. We believe they have improved our report. We have performed some additional experiments to supplement the original version of the manuscript. We believe that both our answers and those additional experiments will be convincing enough and prove our points.

We have therefore added:

- An additional image of Septin bundle in solution visualized by fluorescence microscopy (supl. Fig. 4)
- The distribution of PI(4,5)P2 lipids on curved substrates has been assessed and is displayed in Fig4.A, orange curve. It points out that PI(4,5)P2 lipids are not sensitive to curvature
- Additional liposomes (LUVs) have been prepared and imaged in 2D as control experiments in the absence of protein. Cryo-tomograms of the same control liposomes have also been carried out. The results are displayed in Supplementary figure 7.
- To probe the effect of septins at higher membrane tensions, the stretching modulus has also been determined and displayed in Supl. Fig 3.
- A supplementary figure (1) described schematically the arrangement of Septin oligomers.

Those new findings are described and discussed below.

Reviewers' comments:

Reviewer #1 (Remarks to the Author):

Manuscript ID: NCOMMS-18-20375

We thank referee #1 for his highly positive comments. Please find the answers to his questions and concerns below.

Some comments and questions are below:

(1) Lines 133 – 134 of page 6: Those bundles were most likely self-assembled in solution **prior to their interaction with the vesicles**.

Why can you consider so? What is the specific evidence?

This is a relevant remark. It has previously been shown that when placed in low salt conditions (NaCl concentration < 120 mM) for approximately 30 min, septins can assemble into bundles of filaments (Bertin et al., PNAS, 2008). In our experimental conditions (NaCl = 75 mM for 30 min), septins are thus most likely assembled into bundles, paired filaments and octamers. Indeed as shown in the image below, bundles can be visualized, under the same conditions by fluorescence microscopy (left) and electron microscopy (right).

Hence we have added, in the main text: “Those bundles were most likely self-assembled in solution prior to their interaction with the vesicles **and are visualized, in solution by fluorescence microscopy**”.

*Left: Confocal image of GFP-Septins (170 nM) in 10 mM Tris pH 7.4, 75 mM NaCl. Scale bar: 2 μ m.,
Right: electron microscopy image of a septin bundle.*

(2) Lines 193 and 195 of page 8 (Eqs. 5 and 7):

Please unify the indication style of superscript, multiplier, of n.

The text has been changed accordingly.

(3) Line 240 of page 9:

... (Figure 3 D, left panel, **arrow**).

In the left panel of Figure 3D, the arrow is missing.

We thank the referee for pointing this out. An arrow has been added accordingly.

(4) Lines 242 - 243 of page 9:

Overall, these observations demonstrate that septins deform and constrain small positively curved vesicles (diameters from 100 nm to 1 μ m) into **flat “pancakes”**, ...

It is difficult for me to understand the result indicating that small positively curved vesicles deformed into the flat pancakes. If I look at which part of which Figure, can I understand it?

I hope a dedicated Figure to show that morphology.

We understand that the referee could have been puzzled by the fact that septins dislike interacting with positively curved surfaces while they obviously bind to LUVS (50 nm -1 μ m in diameter). Even though LUVs display a high positive curvature not favorable for Septin interaction, the presence of phosphoinositides enhances the interaction of Septin filaments with those strongly curved liposomes. Bridges et al. (J. Cell Biol., 2016), have indeed shown that septins interact with silica beads of similar diameters covered with a lipid bilayer.

After interacting, the septins tend to impose a more favorable curvature. Since liposomes are obviously too small, they are deformed into those so called “flat pancakes” where septin filaments can lie flat. The morphology of those deformed vesicles is more visible within the supplementary movie 1 in “side views”. Hence a side view of the “pancakes” has been added to illustrate the deformations observed (Fig 3.C 3rd line).

(5) Lines 259 -261 of page 10:

Figure 4.B shows fluorescence intensity measurements for the whole set of templates (Nexperiments = 15, Nimage stacks = 202). As expected, the density of lipids remained constant within the whole curvature range. The lipid bilayer is thus **homogeneous on the templates**.

I agree about the density. But can lipid composition be uniform as well?

Especially, since the head group of PIP2 is very large, is there a possibility that the content will be locally different depending on the local curvature?

We thank the referee for pointing out this possibility. Few lipids are known to be sensitive to curvatures. When they are, they are sensitive to nanometric curvatures. In the present work, septins are “sensitive” to micrometric curvatures. We thus tested this hypothesis with additional experiment. We probed the distribution of PI(4,5)P2 on our curved substrates, using fluorescently labeled PI(4,5)P2. As shown on the figure displayed below and in the new figure 4.A (orange curve), the fluorescence intensity of labeled PI(4,5)P2 remains constant whatever the curvature. Hence the PI(4,5)P2 distribution does not seem to depend on micrometric curvature. In addition, for stronger nanometric curvatures, it has been demonstrated by Tsai et al (eLife 2018;7:e37262 DOI: [10.7554/eLife.37262](https://doi.org/10.7554/eLife.37262)) using “tube pulling experiments” that PI(4,5)P2 is not sorted into nanometric tubes. Instead the distribution of PIP2 remains homogeneous on both GUVs and nanometric tubes, meaning that PI(4,5)P2 is not sensitive to small nanometric curvature

As a consequence, the bilayer composition can be considered uniform, at least in term of PI(4,5)P2 lipid. We have slightly modified the text to report this additional finding: “As expected, the density of lipids, **including the density of PI(4,5)P2**, remained constant within the whole curvature range.”

Relative septin density (green) on SLB deposited on the patterned substrates as a function of the curvature. The red curve corresponds to the lipid signal as a control (rhodamine PE) while the orange curve corresponds to the PIP2 fluorescence intensity.

(6) In Figure 4F (page 23):

Although there is one white arrow, the explanation about it is missing.

Thanks for noticing this. We have added an explanation in the legend of figure 4.F

(7) The legend of Figure 1C-E (page 25):

Do the error bars indicate S.D.? (or S.E.?)

All of the error bars, throughout the manuscript, indicate standard deviations. We have added a sentence in the text to specify it: "All of the errors are calculated using standard deviations".

(8) Lines 568 – 569 in the legend of Figure 4F-G (page 26):

The inset shows the angular distribution of pixels at the top (**red**) (positive curvature) and at the bottom (**black**) (negative curvature of the substrate).

Are they purple and green, respectively?

Thanks for noticing this mistake. We have changed the text accordingly.

(9) On the Methods, no reference is cited in some sections about cryo-EM, SUV formation to assemble supported bilayers, SEM, and persist length measurement.

Are there no cited references to serve as reference?

Some seminal references are now cited in the methods section. We cite Zanetti et al. (2013, elife, 2:e00951. doi: 10.7554/eLife.00951) for Cryo-EM of liposomes in vitro. For SUV formation and fusion

on supported lipid bilayer, we now cite Ellenbroek et al. (2011, Biophysical Journal, 101(9), 2178-84). For SEM principles we now cite Svitkina, 2009, methods in molecular biology, 586, 187-206. Finally Isambert et al. (1995, J. Biol. Chem., 270 (19), 11437-44) serves as a reference for persistence length measurements.

(10) Line 157 of the “Persist length measurement” section of the Methods:

The composition of the solution used in the experiment is much different from the others. Especially, it contained divalent cation and methylcellulose.

Is there no problem?

We understand the concern of the referee. For those experiments the buffer used was: 20 mM imidazole-HCl at pH 7.4, 1 mM dithiothreitol, 0.1 mM MgATP, 50 mM KCl and 2 mM MgCl₂, 1 mM Trolox, 2 mM protocatechuic acid, 0.1 μM protocatechuate 3,4- dioxygenase and 0.1% (w/v) Methylcellulose. TIRF experiments require that the sample is constrained at the bottom of the experimental chamber. We thus used methylcellulose which is widely used in the community to constrain the proteins at the bottom of the chambers. In addition, we have used anti-bleaching agents to prevent the protein degradation. Finally, magnesium, at 2 mM, does not alter the self-assembly of budding yeast septins (see Bertin et al., 2008 and Garcia et al., 2012).

(11) Lines 169 and 174 of the “Persist length measurement” section of the Methods:

Fig S5.A and Fig S5.B should be Fig S6.A and Fig S6.B, respectively.

We thank the referee for noticing this mistake which has been modified accordingly.

(12) About Supplementary Figure 1:

It is divided into two panels (cases without and with septin). But, at least to me, it is easier to understand, when it is put together into one panel.

As suggested by the referee, we have merged the graphs to make our point more demonstrative.

(13) As the authors mentioned, septins are guanine-nucleotide-binding proteins.

However, there is no description about the addition of GTP anywhere, and none of the solutions contain GTP.

Is there no problem?

GDP is added during Septin purification (5 μM GDP). This is sufficient to ensure a proper stability of the Septin complex. The exchange of nucleotides is indeed very slow and we have checked over time that septins remain stable even without adding additional nucleotides in excess.

Reviewer #2 (Remarks to the Author):

We thank the referee for his remarks and concerns. As he mentions he is not a Septin expert. Being a cryo-em specialist, he points out at some imaging issues.

The message of our manuscript does not rely on our cryo-EM experiments. The goal of the present work is not to perform structural analysis per se with high resolution imaging of septin arrangement. However, cryo-EM and tomography qualitatively support our message demonstrating that Septin filaments dramatically deform small vesicles. Besides we do visualize both the liposomes and the proteins in those assays, which was our goal. We want to point out, that for those qualitative assays, we have used a standard 200kV Lab6 microscope which was highly sufficient to prove and understand the curvature sensibility of septins.

To support our statements, we have performed additional control experiments. We have prepared control samples of liposomes and display now some of the resulting 2D images. In addition, cryo-tomograms have been collected to show, in 3D that control liposomes, in the absence of septins, are round shaped (supplementary figure 7, A-C).

We have indeed collected data on similar samples using a cutting edge "Titan Krios" microscope to obtain high resolution data. The goal of these experiments is not to study the curvature sensitivity of septins but rather to describe the organization of septins at high resolution bound to membranes. This is a project in itself and is not at all the concern of the present work. This will be the object and focus of a future paper.

I am serving as a technical reviewer for the cryo-electron tomography (cryo-ET) experiments presented in this paper.

The authors use a number of biophysical methods including cryo-ET to describe how septins interact with differentially curved membrane bilayers in vitro. These results are used to develop a model for how septins might promote cell division.

As I am not an expert on septin biology I cannot judge the novelty and the impact of the paper on the field.

It is good practice in the cryo-EM field to deposit reported data for public access after publication. The authors should submit at least one example tomogram in the Electron Microscopy Data Bank (EMDB) and include the accession code in the paper.

This is a relevant remark. We have indeed submitted one of the tomogram to the EMDB. The EMDB code of the data uploaded within the EMDB is EMDB-0321.

There are several aspects regarding the interpretation of the cryo-ET experiments that should be addressed in a revised version of the manuscript.

1) Major concerns:

- Visual/manual filament segmentation is problematic, because visibility of filamentous structures to the human eye in cryo-ET data strongly depends on their orientation with respect to the imaging direction and the tilt axis. In particular, filaments oriented perpendicular to the imaging direction (i.e. in the sample plane) can be typically recognized and traced much better than filaments oriented parallel to the imaging direction. Could the authors try and trace/segment the septin filaments in an automatic fashion to support their manual segmentation?

We agree that because of the missing cone, filaments lying perpendicular to the direction of the electron beam would be more visible than the filaments oriented parallel to the imaging direction.

We have indeed attempted, with our data to perform automated segmentation using Amira. The segmented filaments automatically detected were still orthogonal to the imaging axis. In addition we also obtained a large number of false positive which we have to eventually remove manually. We have thereby decided to carry out our segmentation manually.

- The authors suggest that septin filaments deform vesicles into flattened "pancakes". This is a very problematic conclusion, because isolated vesicular structure always appear significantly flattened in cryo-ET data, most likely because they are squeezed by the surface tension of the water/buffer film during the blotting process that is part of the cryo-EM grid preparation. There are numerous published cryo-ET studies that document this effect. The authors must consequently test and document systematically that the flattening of vesicles they observe in their data indeed also appears in directions not expected due to the effects of the surface tension.

As the referee suggested, the vitrification process within a thin film of solution might partly induce some flattening of the vesicles. To minimize this effect, we have used lacey grids rather than quantifoil grids, on purpose. Indeed, lacey grids are heterogeneous and the size of the holes (from 50 nm to several microns in diameter) can accommodate from small to large vesicles without inducing drastic deformations as visualized on a control image (Figure 3.A, left panel, left insert).

However, in the images that we had selected (figure 3), we clearly see a difference in morphology for the vesicles which have septins bound. For instance, in Figure 3.A (right panel), the "free" liposomes are clearly round shaped and darker because of the thickness of the vesicles while the vesicles covered with a parallel array of filaments are strongly deformed. Specifically, seen in 2D projections, the contour of the vesicles display sharp angles not visualized on naked vesicles. For clarity, we have added a sentence in the text and arrows in figure 3.: "Liposomes can be obviously deformed by the surface tension imposed by the vitrified film of liquid. However, as visualized in 2D projections (figure 3, right panel), naked vesicles remain perfectly round shaped while decorated vesicles display peculiar morphologies. Indeed sharp angles can be pinpointed on the contour of those vesicles (arrows)."

To support our findings, we have performed additional control experiments. We have prepared control samples of liposomes and display some of the resulting 2D images, shown in supplementary figure 7.A. At both low magnification (suppl 6.A, left panel) and higher magnification (suppl 7.A, right panel) round shaped vesicles are visualized. The inner area of those vesicles appears darker because of their intrinsic thickness. Hence, at higher magnification, only the periphery of a vesicle is displayed (suppl.fig. 7.A, right panel). In addition, cryo-tomograms have been collected to show, in 3D that

control liposomes, in the absence of septins, are round shaped. To this end, additional supplementary movies (movies S3 and S4) and images (Supplementary figure 7.B and C) have been added, displaying the segmentation of naked vesicles.

A. Control image of naked LUVs at low magnification (left, scale bar: 1 μm) and higher magnification (right, scale bar: 100 nm). B and C. slices in tomograms carried out on control samples (left). The membrane segmentation is shown in yellow (right). C: scale bar 50 nm, D: scale bar 100 nm.

2) Minor points:

2.1) Abstract:

- line 15 and several additional instances: "inner plasma membrane" should probably be "inner face"

of the plasma membrane", or "cytoplasmic face of the plasma membrane"

- "septin filament curvature arrangement preferences" is very complicated. Please provide more comprehensive description.

The manuscript has been changed accordingly. "inner plasma membrane" has been replaced by inner face of the plasma membrane". "septin filament curvature arrangement preferences" has been replaced by preferential arrangement of Septin filaments on specific curvatures

2.2) Introduction:

- I think there are several aspects in the paper that could benefit greatly from visualization in a Figure:

* basic palindromic layout of septin filaments (line 40)

* architecture of minimal septin oligomers (line 42)

* geometry of bud necks in dividing cells and septin filament arrangement described in the third introduction paragraph (lines 46-53)

- line 61: "proteins can also form a coat interacting with the membrane TO induce membrane deformation"

A supplementary figure 1.A has been added to enhance clarity. The basic palindromic architecture of Septin octameric complexes has been schematically represented along with their assembly into non polar paired filaments. The geometry of the bud neck was already displayed in figure 5.

Schematic representation of an octameric Septin rod in solution (left). Septins are arranged in a palindromic fashion and self-assemble into non polar paired filaments (right).

2.3) Results:

- line 93: what is the physiological septin concentration? Do the concentrations used in the experiments recapitulate the physiological concentration range?

The physiological Septin concentration is unknown. However, it can vary and septins might be densely concentrated locally at the bud neck, during cell division.

- line 104: "septins.μm-2" should be "septins/μm-2"

We are referring to the density of septins by μm^2 . It is thus septins/ μm^2 which is usually written as septins. μm^{-2}

- line 146-148: has it been shown previously that electrostatic interactions do not play a role anymore at > 150 mM NaCl?

The referee is right. Pure electrostatic interactions are screened at salt concentration > 150 mM. However the PI(4,5)P2 – Septin interaction involves specific interactions in addition to pure electrostatics. For instance, on lipid monolayers and in high salt conditions, the organization and polymerization of Septin filaments is facilitated (see Bertin et al., JMB, 2010).

- Fig. 3A: please enlarge the inserts. They are almost impossible to recognize in a printed version. For clarity, the inserts have been displayed as well in supplementary figure 7.D, where they are more visible than in the original figure. Those inserts only display the membrane morphological modifications.

2.4) Methods:

- line 89: please add the total electron dose of 2D projections.

The 2D projections were collected at a dose of 10 electrons per \AA^2 . We have added this experimental aspect in the text.

- line 93: "tilted series" should be "tilt series"

We have changed "tilted series" for "tilt series" as suggested by the referee.

- line 93: please specify the tilt scheme: were the tilt series acquired in one go from -60° to 60° , was it a bi-directional scheme starting at some low tilt angle, or were the authors using the dose-symmetric "Hagen scheme"?

Since cryo-tomograms have been collected using a side-entry 200kV Lab6 tecnai G2 microscope, we cannot carry out the so called "Hagen scheme", for mechanical stability. However to prevent dramatic irradiations at the lowest tilt angle, we have used an alternative approach. Images are collected first at low angles and finally at the highest tilts following this empirical scheme: 0 to -34 degrees, then +2 to 60 degrees and finally -36 to -60 degrees. A sentence has been added in the manuscript to specify the methods used for tilt series data collection.

- line 95: the authors claim that they acquired tilt images with a constant dose of 0.8 e-/A2 for each

tilt image. This is rather unusual, because exposure is normally adjusted to the increasing sample thickness upon tilting, e.g. in a cosine like manner.

The dose is set at 0.8 electrons per Å^2 at the lowest tilt angles and might indeed be lower at the highest tilts. However, adjusting the dose with tilt angles is usually performed with cutting edge microscopes. We believe that the outcome is not different on our case, using a Lab6 microscope without energy filter.

- line 95: what is the cumulative dose used for acquisition of the entire tilt series?

The total dose is about 100 electrons per Å^2 .

- line 231: please describe how the statistical analysis of filament arrangement based on the cryo-ET data was done.

The error is given by the standard deviation. A sentence has been added in the text accordingly.

- line 234: please visualize schematically the arrangement described here

We have added a schematic representation for clarity in supplementary 1.B, along with the other schemes, presenting the octameric rod and paired filament schematically.

Arrangement of Septin filaments bound to a membrane. To maintain a regular spacing in between filaments, imposed by coiled coils, the octamers might be oriented differently within the same filament.

- line 240: the authors observe filaments rupturing the vesicles. Do they expect this to happen in vivo? If so, what would be the implications?

This observation results from the membrane deformations imposed by Septin filaments. This might

occur in vivo as suggested since septins are multi-tasking proteins. Nonetheless, that would be speculative at this stage to infer a mechanism in vivo.

Reviewer #3 (Remarks to the Author):

We thank referee #3 for suggesting relevant experiments and for his comments. We believe that we have used a synergy of complementary experiments and theory which demonstrate our statements. To enhance our demonstration, we have included additional experiments which will hopefully clarify our point.

Here are some points of concern:

1. Polymerization free energy of septin can be curvature dependent. Energy of bending septins can also penalize the total energy. Both factors are not accounted for in the model

Unlike stated by the referee, the bending energy of septins is included in the model and corresponds to the terms with the persistence length, L_p , in the equations.

The model does not include the actual polymerization process of septins and assumes that they are already in the form of filaments. Indeed, we argue in the discussion that the curvature sensitivity of septins is a property of their filamentous nature and not the result of a guided polymerization process, as the curvature dependence on the orientation of bound octamers is negligible compared to the rotational degree of liberty of octamers, see discussion. In any case, the polymerization free energy of septin filaments **should not dependent on the sign of curvature**, which is only defined relative to the bound membrane. Therefore, the curvature dependence of septin polymerization free energy, if it exists, could not explain the sensitivity of septin filaments to the sign of curvature when binding to membranes.

2. Are entropic terms -- conformational entropy of septins or the membrane important? How can the authors rule this out?

Entropic terms are included in the measured free energy parameters (for instance the binding free energy of septin on membranes).

The free energy cost of deforming the membrane is neglected in the model, as it is expected to be significantly smaller than the free energy cost of deforming septin filaments. The cost of deforming a $1 \mu\text{m}^2$ patch of membrane can be estimated as $\kappa \Delta c^2$ where $\kappa = \sim 10 \text{ kT}$, which is comparable with the cost of deforming only a few septin filaments bound to the $1 \mu\text{m}^2$ patch of membrane: $kT L_p \Delta c^2$ where $L_p = 2L = 2 \mu\text{m}$, for each single filament.

3. Does PIP2 distribution on the membrane depend on curvature? The current model assumes it is not. If so the curvature dependence of septin adhesion can be explained using PIP2 spatial distribution coupled to curvature.

This relevant remark corroborates with point 5 raised by referee #1. We have performed additional experiments which demonstrate that the distribution of PIP2 is independent on the curvature in the μm range (see above and new figure 4.A). In addition, for stronger nanometric curvatures, it has been demonstrated by Tsai et al (eLife 2018;7:e37262 DOI: [10.7554/eLife.37262](https://doi.org/10.7554/eLife.37262)) using “tube pulling experiments” that PIP2 is not sensitive to nanometric curvatures.

4. It would be more compelling if the authors performed experiments at different PIP2 concentrations and also develop a model that can account for this PIP2 dependence.

We have recently carried out a comprehensive methodological study on analyzing and enhancing the Septin-PIP2 interaction on membranes. In this report (Beber et al, in press, cytoskeleton, <https://doi.org/10.1002/cm.21480>), we have probed different concentrations of PI(4,5)P2 on supported bilayers and found that the density of septins plateaus starting at 5-6 % of PI(4,5)P2 (w/w) incorporated in the GUV membrane. In the present manuscript, the PIP2 concentration ranges is above 5%. At the plateau, membranes are thus saturated by PI(4,5)P2 and at equilibrium are saturated as well with septins (see suppl fig. 5)

5. The authors can also try and manipulate the spatial organization of PIP2 -- say by cholesterol or other means and challenge the mathematical model

Cholesterol is already present in our lipid mixture and as pointed previously the PIP2 distribution is homogeneous in the membrane. It is not really known how PIP2 could be spatially distributed differently on specific micrometric curvatures. Divalent ions as already noticed by others can cluster PIP2. If the distribution of PIP2 is not homogenous but is clustered within domains, uncontrollable parameters would be added to the

6. It is not clear why membrane tension does not contribute to the explanation. Can the authors perform the experiments under different tension? Can the mathematical model be formulated to include tension? Can a comparison between the model and experiment be made in a tension dependent fashion?

The septins reorganization and curvature has been observed on different vesicular systems (ie LUV, GUV and SLB). The membrane tension cannot be controlled and tuned on LUVs and SLB. On SLBs, we clearly show that the organization of septins depends on the curvature.

However, using GUVs, we can tune and control the tension of membranes. We thus assessed whether the presence of septins alters the mechanical properties of membranes. We have seen that imposing a tension using a micro-pipette prevents any macroscopic deformation to occur at 200 nM Septin concentrations. Besides, we have shown that under low tensions, the bending modulus is not affected by the presence of septins (figure 1). To probe the response of membranes using higher tensions we have performed additional experiments. We thus obtained the stretching modulus of the membrane as seen below. Here we notice a mild effect of septins on the stretching modulus of the membrane while no effect could be deciphered at a lower tension regime. The stretching modulus in the presence of septins ($42 \pm 8 \text{ mN/m}$) is slightly lower than the stretching modulus of

naked vesicles (65 ± 9 mN/m). In the presence of septins, the stretching modulus might be lower because the filaments might slightly insert within the membrane.

Applied tension versus area expansion at high tension (>0.5 mN/m). The red curve corresponds to the control and the green curve to the septin-coated experiments. The values extracted from the slopes are $K_{control} = 65 \pm 9$ mN/m and $K_{septins} = 42 \pm 8$ mN/m

Numerical simulations (ref DOI 10.1186/s13628-014-0013-3) have shown that septins function is best conserved when they are shallowly inserted into the membrane (1.2 nm), as well as several EM observations of septin filaments laying on membranes. This suggests that septin insertion between lipids could be triggered by external stretching of the membrane.

Hence, because membrane tension is barely affected by the presence of septins we did not include tension into the model.

To report for those now sets of experiments, we have added an additional supplementary figure (suppl. Fig. 3) and modified the text accordingly.

7. I think by adding extra dimensions to challenge the model against experiment (PIP2, tension) is important to establish unequivocally the explanation for the results. Otherwise the authors are just providing a plausible explanation and not a definitive one.

Indeed, the variation of energy with PIP2 concentration could be included into the model. However, in our experimental conditions, the amount of protein bound to the supported lipid bilayer plateaus and thus the energy of interaction is independent on the PIP2 concentration, in the chosen regime. Similarly, the membrane tension on supported lipid bilayers cannot be tuned. Hence we had decided not to include these additional parameters in our model.

8. Does inter filament interaction contribute to the assembly? Can the authors comment on how or manipulate this?

We do not know of any mutant which could prevent lateral interactions between filaments. However as we have shown, end to end interaction and thus polymerization is essential for septins to

distribute on specific curvatures.

Reviewers' Comments:

Reviewer #1:

Remarks to the Author:

Manuscript ID: NCOMMS-18-20375A

The revised version of the manuscript entitled "Membrane reshaping by micrometric curvature sensitive septin filaments" by Alexandre Beber et al. adequately responded to all my comments and questions.

They are adding as much data and consideration as possible at the current to the revised version, and again I would point out that both the method used and results obtained in this study are very interesting.

Thus, I would like to recommend that the manuscript would be published in Nature Communications.

Some comments are below:

(1) On the legend of Fig. 4B that has been revised to show the distribution of PIP2 on the curved substrates:

Although there is additional data, explanation about the orange curve is missing.

By the way, in the reply mail by the authors, it is written everywhere as Fig4.A (or Figure 4.A). However, correctly, it might be Fig. 4B.

(2) On the phrase of "pancake" (this word is used in two places in the text):

I understand the things the authors would like to describe.

But, I think that this word imagines the reader to a more typical morphology like Nanodisc, causing misunderstanding. It seems better to replace it with another word such as simpler, "flat shape" or "flat morphology".

Reviewer #2:

Remarks to the Author:

The authors have adequately addressed all of my concerns.

Reviewer #3:

Remarks to the Author:

My questions from the previous round are addressed more or less.

We are grateful to the referees for their highly positive comments.

Please find below (highlighted in red) the responses to the minor comments of reviewer 1.

REVIEWERS' COMMENTS:

Reviewer #1 (Remarks to the Author):

Manuscript ID: NCOMMS-18-20375A

The revised version of the manuscript entitled “Membrane reshaping by micrometric curvature sensitive septin filaments” by Alexandre Beber et al. adequately responded to all my comments and questions.

They are adding as much data and consideration as possible at the current to the revised version, and again I would point out that both the method used and results obtained in this study are very interesting.

Thus, I would like to recommend that the manuscript would be published in Nature Communications.

Some comments are below:

(1) On the legend of Fig. 4B that has been revised to show the distribution of PIP2 on the curved substrates:

Although there is additional data, explanation about the orange curve is missing.

By the way, in the reply mail by the authors, it is written everywhere as Fig4.A (or Figure 4.A). However, correctly, it might be Fig. 4B.

Thanks for noticing the mistake. We have changed the legend of figure 4B to describe the displayed orange curve: “The red and orange curves correspond to the lipid signals as controls (Bodipy TR-ceramide and fluorescent PI(4,5)P2 respectively)”.

(2) On the phrase of “pancake” (this word is used in two places in the text):

I understand the things the authors would like to describe.

But, I think that this word imagines the reader to a more typical morphology like Nanodisc, causing misunderstanding. It seems better to replace it with another word such as simpler, “flat shape” or “flat morphology”.

We have removed the inelegant expression “flat pancakes” to rephrase it as advised and described those deformed vesicles as “flattened” for instance.

Reviewer #2 (Remarks to the Author):

The authors have adequately addressed all of my concerns.

Reviewer #3 (Remarks to the Author):

My questions from the previous round are addressed more or less.